# Digital non-Foster-inspired electronics for broadband impedance matching

Xin Yang [1,2,6] ✉, Zhihe Zhang[1,6], Mengwei Xu[1], Shuxun Li[1], Yuanhong Zhang[1], Xue-Feng Zhu [3] ✉, Xiaoping Ouyang[4] ✉ & Andrea Alù [5] ✉

Narrow bandwidths are a general bottleneck for applications relying on passive, linear, subwavelength resonators. In the past decades, several efforts have been devoted to overcoming this challenge, broadening the bandwidth of small resonators by the means of analog non-Foster matching networks for radiators, antennas and metamaterials. However, most non-Foster approaches present challenges in terms of tunability, stability and power limitations. Here, by tuning a subwavelength acoustic transducer with digital non-Foster-inspired electronics, we demonstrate five-fold bandwidth enhancement compared to conventional analog non-Foster matching. Long-distance transmission over airborne acoustic channels, with approximately three orders of magnitude increase in power level, validates the performance of the proposed approach. We also demonstrate convenient reconfigurability of our non-Foster-inspired electronics. This implementation provides a viable solution to enhance the bandwidth of sub-wavelength resonance-based systems, extendable to the electromagnetic domain, and enables the practical implementation of airborne and underwater acoustic radiators.

Resonance-based systems, including acoustic radiators, electromagnetic antennas, and metamaterials, suffer from a well-known trade-off between bandwidth and size[1], because of passivity and causality constraints[2,3]. A trade-off between efficiency, bandwidth, and electrical size of resonance-based systems plagues the implementation of many devices and technologies. In the context of compact acoustic radiators and electromagnetic antennas, the well-established trade-off between bandwidth and size is known as Chu's limit[4,5]. Similarly, a general constraint between the operational bandwidth and the device size exists in acoustic and microwave metamaterials[6]. These bounds are the key benchmarks for resonance-based system design[6,7]. Widening the bandwidth beyond these bounds is possible only by breaking one or more of the underlying assumptions on time invariance, passivity, and linearity[3,4].

Temporal modulations of antenna elements or the connected matching circuitry have been proven to be effective in this context[1,8,9]. Breaking passivity to achieve a wide bandwidth was recently implemented by loading small antennas with analog non-Foster circuitry[10,11], commonly implemented by an analog negative impedance converter that violates Foster's reactance theorem[12,13]. Recently, this circuitry was used to enhance acoustic radiation from a piezoelectric transducer, which successfully surpassed the acoustic Chu's limit[4]. Similar approaches enable parity-time symmetric electronics[12,13] and acoustics[14,15]. Analog non-Foster circuitry has also been explored for broadband microwave generation[16], robust wireless power transfer[17,18], nonreciprocal transmission of microwave acoustic waves[19], as well as broadband unidirectional acoustic devices[20].

[1]College of Electrical and Information Engineering, Hunan University, Changsha 410082, People's Republic of China. [2]Engineering Research Center of Advanced Semiconductor Technology and Application of Ministry of Education, Hunan University, Changsha 410082, People's Republic of China. [3]School of Physics, Huazhong University of Science and Technology, Wuhan, Hubei 430074, People's Republic of China. [4]School of Materials Science and Engineering, Xiangtan University, Xiangtan 411105, People's Republic of China. [5]Photonics Initiative, Advanced Science Research Center, City University of New York, New York, NY 10031, USA. [6]These authors contributed equally: Xin Yang, Zhihe Zhang. ✉e-mail: xyang@hnu.edu.cn; xfzhu@hust.edu.cn; oyxp2003@aliyun.com; aalu@gc.cuny.edu

However, analog non-Foster circuits are limited in their available parameter space and hence are unable to engineer arbitrary frequency dispersion and provide real-time reconfigurability. They can be configured in a discrete and constrained manner due to the limited choice of available circuit components. Once the circuit structure is fixed, the negative impedance cannot be adjusted in operation or be altered. Delicate manual tuning of variable circuit components may discourage their broad applicability[12,18]. In addition, power handling[21] and stability limitations[22,23] of analog non-Foster circuits hinder the implementation of a broad range of metamaterial devices in which bandwidth is important. These shortcomings plague a diverse range of practical applications.

In the past decades, several research prospects have been put forward to overcome these challenges. The concept of digital meta-materials has been introduced to provide metamaterials with robust reconfigurability and modularity[24]. Active control can add degrees of freedom to tune the metamaterial parameters[25,26] and achieve gate-tunable negative refraction[27]. These features can be successfully used to realize broadband sound absorbers[26] or sound barriers[28]. Recent breakthroughs have also been made in the context of programmable mechanical metamaterials[29,30]. Virtualized metamaterials allow us to freely reconfigure the frequency dispersion (FD) using a digital representation of tunability solely based on software modifications[31]. The pioneering digital non-Foster circuits for realizing negative impedance are based on open-loop control, which in fact mimic ordinary analogy non-foster systems by specific functions of digital filters[32–36]. The concept of using additional sources that simultaneously implement non-Foster cancellation and negative resistance was introduced[37,38]. A high energy-injection switch-mode amplifier has been shown to realize a negative inductance[39] and a nonlinear negative resistance[40], showcasing low loss compared to analog operational amplifiers (op-amps)[17,41].

In this work, inspired by these works, we demonstrate the realization of digital non-Foster electronics with tailored and reconfigurable FD. Our digital non-Foster-inspired circuitry enables a broad control over the amplitude-phase relation between the terminal voltage and current, which can be set arbitrarily, hence synthesizing in real-time an equivalent negative resistance, negative inductance or negative capacitance with on-demand FD. A self-adaptive proportional-resonant (PR) controller[42,43], which is pre-programed in a digital microprocessor, can actively select the optimal control characteristics based on the operating frequency to ensure excellent transient response and real-time tunability. More importantly, this solution successfully circumvents inherent instabilities of non-Foster circuits by ensuring that the poles of the characteristic equation remain in the stable portion of the complex plane for a wide range of control parameters. As an additional advantage, the op-amps used in analog non-Foster circuits have a limited range of power handling, while the use of switch-mode electronics may provide a leap forward also in power levels. Based on this platform, we demonstrate that loading a low-resonance-frequency electroacoustic transmitter with our digital non-Foster-inspired electronics supports broadband high-power acoustic radiation. Our simulation and experimental results show that the proposed digital non-Foster electronics offer flexibly engineered FD, enhancing the operation bandwidth by over five times compared to a matching network based on analog non-Foster electronics. In experiments, we apply the digital non-Foster electronic network to demonstrate long-haul image transmission over airborne acoustic channels, validating the stability of the proposed system, and demonstrating its flexibility, reconfigurability, and real-time tunability.

## Results
### Digital non-Foster-inspired electronics
The key element of conventional analog non-Foster circuitry (Fig. 1a) is the op-amp. Since the supply voltage is limited to tens of volts in analog devices, this topology is impractical for high-voltage scenarios. The op-amp realizes a negative impedance, crucial for the classic implementation of non-Foster matching circuitry[9] (Fig. 1b). However, constrained by the commercially available component values and tolerances of passive analog RLC elements, the negative impedance attainable with analog non-Foster circuits cannot span a continuous range, and their tuning often depends on tedious circuit component replacements. For example, capacitance specifications for common passive capacitor packages only include 100 pF, 220 pF, 3.3nF, 6.8nF, etc., with the tolerances of up to ±20%[44]. Therefore, analog non-Foster matching networks cannot synthesize arbitrary FD, facing challenges with discretized parameter space and complicated manual tuning by tedious circuit component replacements. Power limitations and limited dispersion engineering leaves the full bandwidth-enhancement potential of non-Foster matching untapped.

Several efforts have been devoted to overcome these obstacles. Digital non-Foster electronics (Fig. 1c) enables opportunities in this context. The transition from analog to digital circuits provides more degrees of freedom in the reconfiguration and tunability of equivalent negative impedance over a wide parameter range[31]. Digital micro-processors allow operations exempt from delicate manual tuning (i.e. change the circuit RLC elements to tune the negative impedance set by analogue nonfoster circuits), altering the negative impedance reference and controller parameters on-demand through software-defined programming. Meanwhile, switch-mode electronics can lead to a leap in available power levels[40,41]. In stark contrast with op-amp-based analog non-Foster electronics, the proposed digital non-Foster-inspired electronics is characterized by arbitrary FD with equivalent negative resistance and inductance/capacitance, exploiting a feedback control to actively configure the amplitude-phase relation of the terminal output voltage $u_o$ and current $i_o$ (Fig. 1d) as necessary.

### Theoretical analysis of digital non-Foster-inspired electronics
A common electroacoustic transmitter, typically operating near a resonance, supports a high-quality factor with limited operational bandwidth[45]. Here, we consider a transducer whose input terminal impedance shows a strong frequency dependence due to eddy currents. By loading it with the proposed digital non-Foster circuit, we demonstrate high-power acoustic broadband radiation. The transducer bandwidth is defined by the half-power level of the acoustic radiation[46]. The acoustic power $P_{ar}$ can be calculated using[47]

$$\begin{cases} P_{ar} = \frac{1}{4}|V_D|^2 \text{real}(Z_{rad}) \\ Df = f_h - f_l \end{cases} \tag{1}$$

where $V_D$ is volume velocity varying with the transducer input voltage, and $Z_{rad}$ is the acoustic radiation impedance. The frequencies $f_l$ and $f_h$ are the half-power frequencies, thereby determining the bandwidth $\Delta f$. According to the overall system impedance model (Fig. 2a), when the transducer is loaded with the digital non-Foster circuit, $V_D$ can be derived as

$$V_D = \frac{BlS_D u_E}{(Bl)^2 + (Z_m + Z_{rad}S_D^2)(R_D(\omega) + j\omega L_D(\omega) + Z'_E(\omega) + R_E)} \tag{2}$$

where the total impedance $Z_D(\omega)$ consists of an inductive component $L_D(\omega)$ and a resistive component $R_D(\omega)$ (see Supplementary Note 1 and Supplementary Fig. 1). The eddy current impedance $Z'_E(\omega)$ comprises the frequency-dependent elements $R_E(\omega)$ and $L_E(\omega)$, which can be modeled as

$$\begin{cases} Z'_E(\omega) = R'_E(\omega) + j\omega L_E(\omega) = L_{ex}\omega^p e^{j\theta(\omega)}, 0 < p < 1 \\ \theta(\omega) = \frac{\pi}{2}\omega^{-q}, 0 < q < 1 \end{cases} \tag{3}$$

where $L_{ex}$, $p$ and $q$ are model parameters[48].

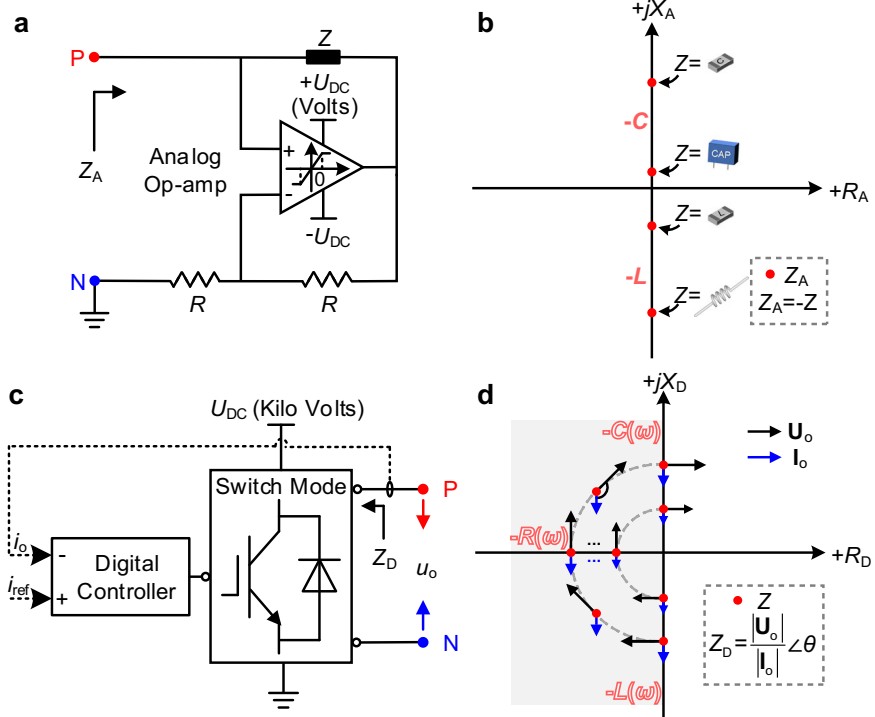

**Fig. 1 | Properties of analog non-Foster circuitry and our proposed digital non-Foster-inspired electronics. a** Analog non-Foster circuitry using op-amps. By changing the passive analog loads $Z$, the negative impedance $Z_A$ can be realized. **b** Diagram of negative impedances with the analogue non-Foster circuit by varying discrete analog components $-Z$ ($-L$ or $-C$). **c** The proposed digital non-Foster-inspired electronics composed of a digital controller and switch-mode circuit adopts a current closed-loop feedback control where $i_{ref}$ is calculated by a reference negative impedance. The software-defined implementation circumvents the inherent limited selection of analog passive elements in analog non-Foster circuits, and enables arbitrary FD and real-time tunability. **d** Actively regulated amplitude-phase relation between $U_o$ and $I_o$, which can continuously realize an equivalent negative resistance $R_D(\omega)$, negative inductance $L_D(\omega)$, or negative capacitance $C_D(\omega)$.

For a low-frequency transducer, the impedance of the mechanical part is much smaller than the electrical clamped impedance at frequencies far from the resonance[45]. Moreover, as the frequency $\omega$ goes up, the eddy current impedance $Z'_E(\omega)$ significantly increases and it becomes the main component of the transducer impedance. More eddy losses will further reduce the acoustic power. This results in a reduction in radiation efficiency and considerably limits the operational bandwidth.

We now explore the relationship between $Z_D(\omega)$ and $\Delta f$ to demonstrate the necessity of utilizing digital non-Foster electronics with arbitrary FD to broaden the high-power transducer bandwidth (Fig. 2b). The outlined black curve shows the limited improvement of conventional analog non-Foster electronics without FD engineering on the bandwidth when $R_D$ is equal to zero. Under this circumstance, the realizable bandwidth enhancement is represented by $p_4$[4]. Complete cancellation of $L_E(\omega)$ ($p_3$ in Fig. 2b) using the proposed negative impedance achieves a bandwidth enhancement. However, it is still limited by the eddy current loss resistance $R_E(\omega)$, which can be seen by the significant difference in bandwidth enhancement between $p_2$ and $p_4$ in Fig. 2b. Both $R_E(\omega)$ and $L_E(\omega)$ vary significantly with frequency. Cancelling out $R_E(\omega)$ and $L_E(\omega)$ over a broad frequency range, $p_1$ further broadens the bandwidth based on the configuration $p_2$. Accordingly, the proposed digital non-Foster element can overcome the performance challenges of conventional analog non-Foster circuits and broaden the operational bandwidth.

Figure 2c highlights the difference between non-Foster electronics with and without FD. The proposed digital non-Foster-inspired electronics can offset $R'_E(\omega)$ and $L_E(\omega)$ across a much larger bandwidth than the analog circuitry. The total resistance of the circuit is almost constant, while the reactance is almost eliminated across a wide bandwidth. The acoustic radiated power for both the proposed non-Foster electronics with and without FD are shown in Fig. 2d. Similar to our analytical calculations ($p_1$ and $p_4$) in Fig. 2b, the bandwidth expansion of the proposed digital non-Foster electronics with FD is over five times larger than the analog scenario.

## Implementation of digital non-Foster-inspired electronics

The general structure diagram (Fig. 3a) and software control (Fig. 3b) jointly ensure the functionality of the digital non-Foster-inspired electronics. Its features are rooted in the fact that the amplitude-phase relation between the terminal output voltage of our non-Foster-inspired circuit $u_o$ and the terminal current $i_o$ can be flexibly adjusted according to the negative impedance reference $Z_{ref}$ through a current closed-loop feedback control, ultimately providing the desired dispersion of the terminal impedance $Z_D$.

By loading the electroacoustic transducer with the proposed digital non-Foster-inspired electronics (Fig. 3a), the power amplifier output voltage $u_s$ and current $i_o$ are first converted by the sensors-based signal conditioning circuit into the voltage signals, which meet the input voltage range of the analog-digital conversion (ADC) sampling, and ultimately are supplied to the digital signal processor (DSP).

After initializing peripherals, the DSP remains in a waiting state until the eCAP interrupt service routine (ISR) or the timer ISR occurs (Fig. 3b). The eCAP ISR completes the calculation of operating frequency $f_s$ by detecting the rising edges and falling edges. In the timer ISR, the reference current $i_{ref}$ is first calculated, which is equal to the power amplifier voltage $u_s$ divided by a target negative impedance $Z_{ref}$ plus the transducer impedance $Z_T$. Before the self-adaptive PR controller is introduced, the measured current $i_o$ has a significant deviation from $i_{ref}$ (Fig. 3c). Thereafter, the adaptive PR controller is used to

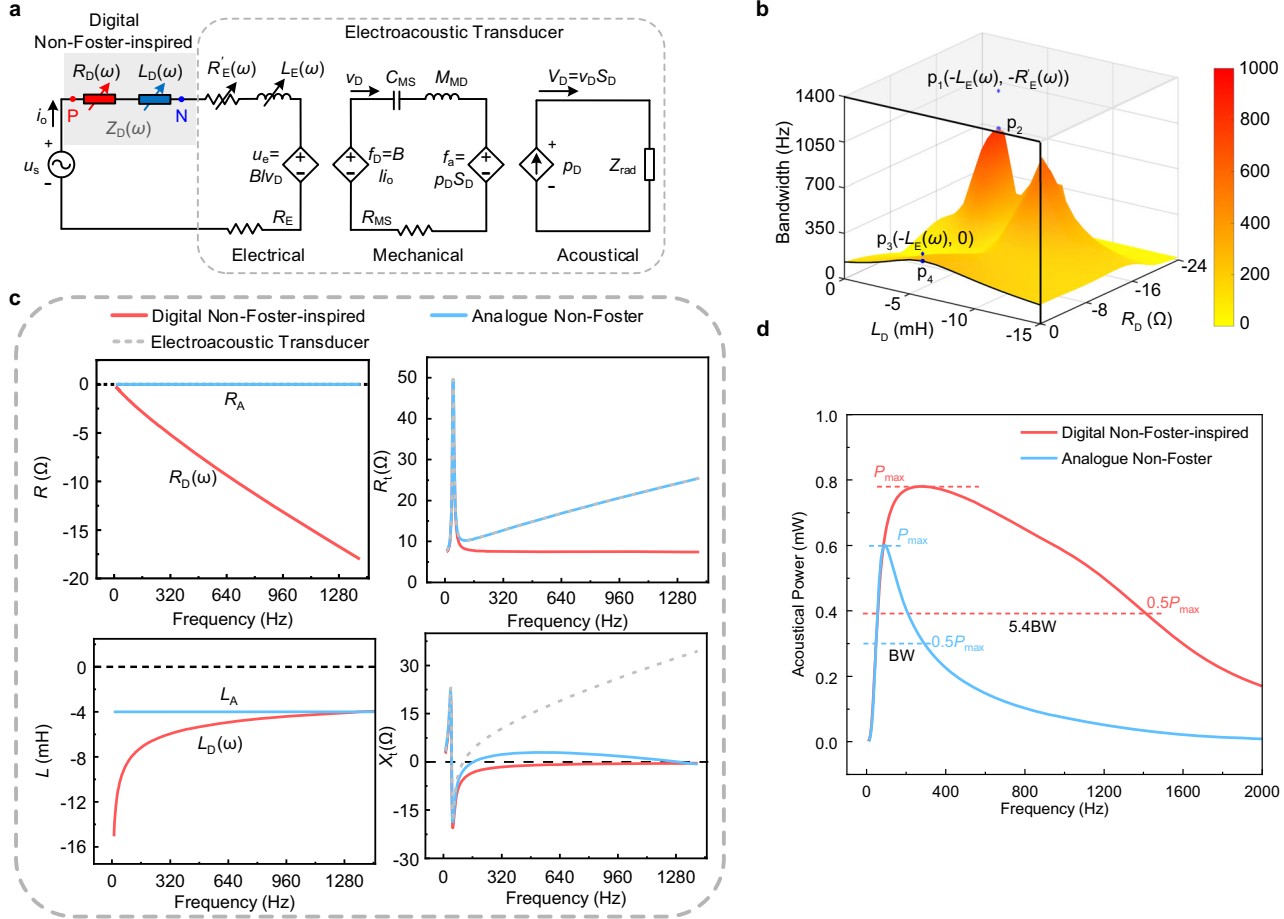

**Fig. 2 | Analysis and design of digital non-Foster-inspired electronics. a** The overall system impedance model diagram. $L_D(\omega)$ and $R_D(\omega)$ at the output terminal impedance of non-Foster circuit. In the ideal lossless analog non-Foster circuit, $L_D(\omega)$ is independent of frequency and $R_D(\omega)$ is equal to zero. In our digital non-Foster-inspired circuit, $L_D(\omega)$ or $R_D(\omega)$ vary with frequency, tracking the transducer dispersion. The rest of the circuit model shows the electroacoustic transducer, with its electrical part, mechanical part and acoustic part. **b** Transducer bandwidth analysis by the analog circuit model after impedance matching using different configurations of $L_D(\omega)$ and $R_D(\omega)$. The entire curved surface represents the bandwidth enhancement without FD. The outlined black curve represents the bandwidth enhancement when $R_D(\omega)$ is equal to zero. Herein, $p_4$ is the maximum bandwidth enhancement that could be achieved[4]. $p_2$ is the maximum bandwidth enhancement on the black surface. $p_1$ is the bandwidth improvement when $R_D(\omega)$ and $L_D(\omega)$ completely offset $R'_E(\omega)$ and $L_E(\omega)$, respectively. $p_3$ is the bandwidth improvement when only $L_D(\omega)$ is offset by $L_E(\omega)$ and $R_D(\omega)$ is equal to zero. **c** Negative impedance characteristics of analog non-Foster and digital non-Foster-inspired electronics and the overall system impedance after impedance matching. **d** Acoustic power curves derived by loading the transducer with analog non-Foster and digital non-Foster-inspired electronics, respectively.

minimize the sinusoidal steady-state errors by a large gain, so that the target negative impedance can be realized. This is achieved by comparing the controller output signal with the triangular carrier wave to generate the pulse width modulation (PWM) signals to control $S_1 \sim S_4$ (Fig. 3d)[49]. By setting the negative impedance reference $Z_{ref}$ into the DSP, the output behavior of the DSP is then established through the current closed-loop feedback control.

The gate driver amplifies the DSP output signal and drives the action of H-bridge converter (Fig. 3e). By the self-adaptive PR approach, our proposed digital non-Foster-inspired circuit provides feedback control on the voltage-current output characteristics (Fig. 3f), which can stably exhibit an equivalent negative impedance at an arbitrary operating frequency $f_s$, but also has a fast dynamic performance. Compared with analog non-Foster circuits that are prone to instabilities[4,50], the proposed digital non-Foster circuit has a wider stability range (see Supplementary Note 2 and Supplementary Fig. 2), because of the adaptive PR controller, which can only be implemented digitally[51] (see Supplementary Note 3 and Supplementary Fig. 3). The desired amplitude-phase relationship between $u_o$ and $i_o$ can be then

achieved to match the required FD to broaden the operational bandwidth.

## Experimental verification

Based on the general structure diagram (Fig. 3a), Fig. 4a illustrates the hardware implementation of digital non-Foster-inspired electronics. The part numbers or values of the main components is shown (Supplementary Tables 1-3). We connected the digital non-Foster-inspired circuit to the electroacoustic transducer to verify the bandwidth-enhancement performance (Fig. 4b). Here, a laser sensor was used to measure the vibration speed of the transducer, which is then converted into the corresponding SPL (see Supplementary Note 4 and Supplementary Fig. 4). An oscilloscope was used to measure the voltage and current waveforms in real time, while the power analyzer quantified the equivalent frequency-dependent negative resistance and inductance, e.g., $R_D(\omega)$ and $L_D(\omega)$.

Due to the programmable control, the designed circuit can synthesize a wide range of equivalent impedance values. The steady-state impedance of both scenarios, with and without FD, are shown in

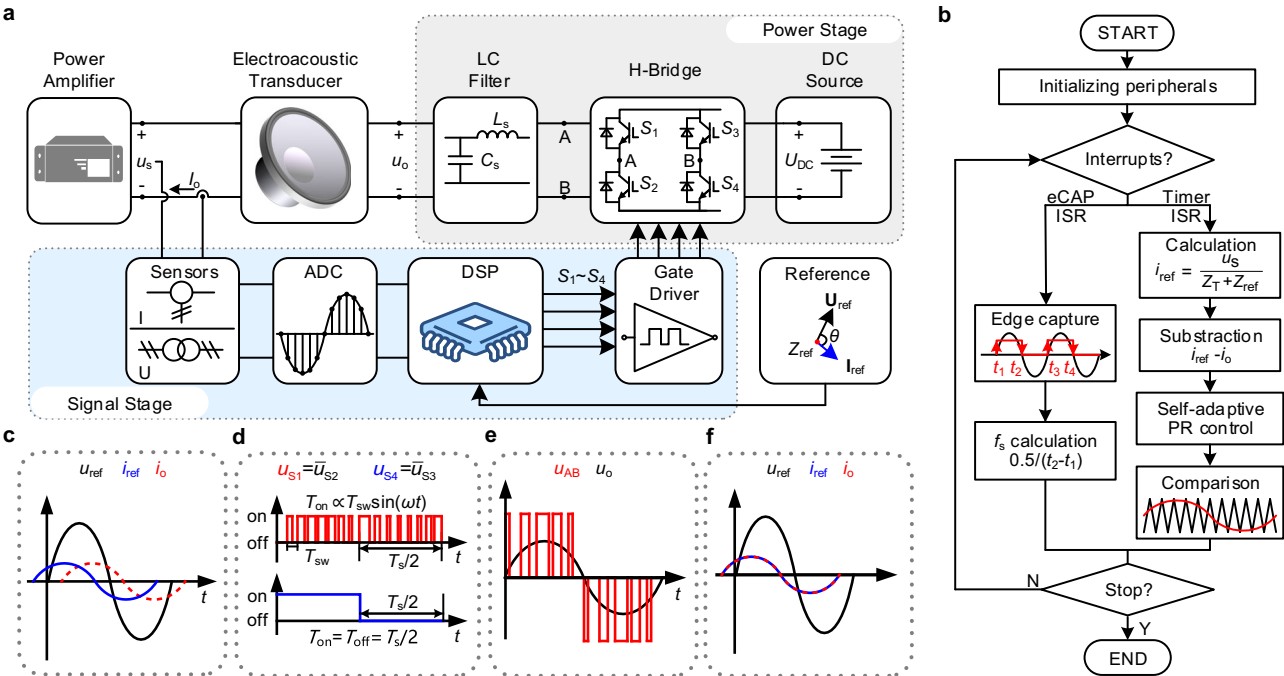

**Fig. 3 | Implementation of digital non-Foster-inspired electronics. a** General structure diagram of the digital non-Foster-inspired electronics. The proposed electronics consists of the signal stage and the power stage. The signal stage contains the signal conditioning (sensors), signal sampling (ADC), DSP, signal output (gate driver). The amplitude-phase relation of the reference current $i_{ref}$ and the reference output voltage $u_{ref}$ of the proposed circuit is determined by setting the negative impedance reference $Z_{ref}$ into the DSP. The power stage includes a DC source, a H-bridge with four SiC MOSFETs ($S_1$ - $S_4$), and a LC filter formed by inductor $L_s$ and capacitor $C_s$. **b** The control program flowchart in DSP, including initializing peripherals, eCAP ISR for the operating frequency $f_s$ calculation and timer ISR for control signal generation. **c** Waveform schematic of the reference voltage $u_{ref}$, the reference current $i_{ref}$, and the output current $i_o$ before the usage of self-adaptive PR closed-loop feedback control. **d** The output signal $u_{S1}$ - $u_{S4}$ of DSP, which is used to drive the behavior of the switch-mode electronics $S_1$ - $S_4$. **e** Output voltage waveform schematic of H-bridge $u_{AB}$ and LC filter $u_o$. **f** Waveform schematic of $u_{ref}$, $i_{ref}$, and $i_o$ after closed-loop control.

Fig. 4c. As expected, the negative impedance provided by the digital non-Foster-inspired circuit is able to offset $R_E(\omega)$ and $L_E(\omega)$ over a broad bandwidth. When the designed circuit presents a frequency-independent resistance and inductance like in conventional analog non-Foster electronics, $R_D(\omega)$ and $L_D(\omega)$ are constant and the corresponding bandwidth shrinks. Please note that conventional analog non-Foster circuits cannot realize the presented negative impedance characteristics here, due to limited power handling of the op-amps. Therefore, the designed circuit has great versatility, even without considering its FD engineering features. The impedance error between theoretical analysis and practical implementation is shown (Supplementary Note 5 and Supplementary Fig. 5).

Figure 4d shows the bandwidth expansion of the transducer under different negative impedance matching approaches. In Fig. 4d, the passive matching using conventional Foster components is shown[4,45]. Here, the total capacitance value is equal to 1000 μF, which is used to compensate the eddy current inductance of the acoustic radiator. The achieved resonant frequency is 50.98 Hz with a bandwidth of approximately 137 Hz. After offsetting the eddy current impedance $Z_E(\omega)$ via the proposed digital non-Foster circuit, the system bandwidth is more than five times larger than the one of conventional analog non-Foster matching, and about eight times the one of passive matching.

## Scenarios for high-power broadband acoustic radiation
The bandwidth expansion realized by the digital non-Foster electronics can synthesize negative resistance and inductance with engineered FD. Its enhanced performance benefits from self-adaptive reconfigurability and real-time control. In order to enhance the tunability to different frequency regimes, the amplitude- and phase-

frequency characteristics of the PR controller are divided into several intervals (Fig. 5a). The controller can automatically capture $f_s$ and alter the controller parameters.

Figure 5b shows the transient characteristics for different frequency spans. When $f_s$ changes, the proposed digital non-Foster circuit can be quickly stabilized and the transient process lasts only two or three cycles, which can be further optimized to approximately within one cycle by improvement of the control algorithm (see Supplementary Note 7 and Supplementary Fig. 7). The distinct phase relation of $i_o$ being ahead of $u_o$ shows a negative-inductance behavior. Since the phase difference is not exactly 90 degrees, the terminal impedance of the circuit also supports an equivalent negative resistance. Moreover, the power analyzer displays the values of $R_D(\omega)$ (UDF$_5$) and $\omega L_D(\omega)$ (UDF$_3$), which exhibit tailored frequency dependence.

It can also be seen in Fig. 5b that the digital non-Foster circuit can handle large voltages and currents (see Supplementary Note 6 and Supplementary Fig. 6). These performance metrics broaden the application scenario of digital non-Foster electronics. Here we apply this technique to transmit an image via airborne acoustic channels, to realize broadband and long-haul communication based on high-power acoustic radiation (Fig. 5c). Due to rapid tracking of different tone frequencies with optimal control characteristics, audio signals encoded via the frequency shift keying (FSK) modulation[52] can be loaded continuously and stably onto the electroacoustic transmitter matched through the designed digital non-Foster circuit. To distinguish the bandwidths of the electroacoustic transmitter under different impedance matching approaches, here yellow, red, blue, black, and white are represented by 200 Hz tone, 300 Hz tone, 600 Hz tone, 1000 Hz tone, and 1200 Hz tone, respectively (see Supplementary Note 8 and Supplementary Figs. 8-9). Since the electroacoustic transmitter

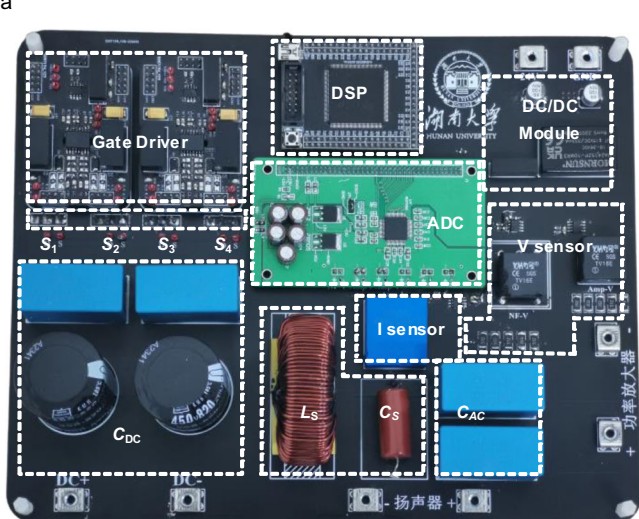

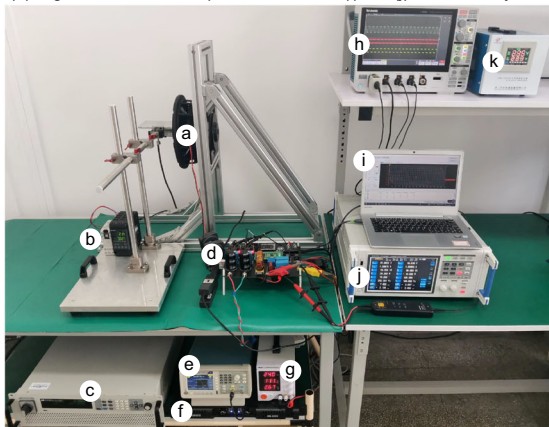

(a) Transducer (b) Laser Sensor (h) Oscilloscope (k) Transformer
(d) Digital Non-Foster-inspired Electronics (i) PC (j) Power Analyzer

(e) Signal Generator (g) Auxiliary Power Supplier (c) DC Source
(f) Power Amplifier

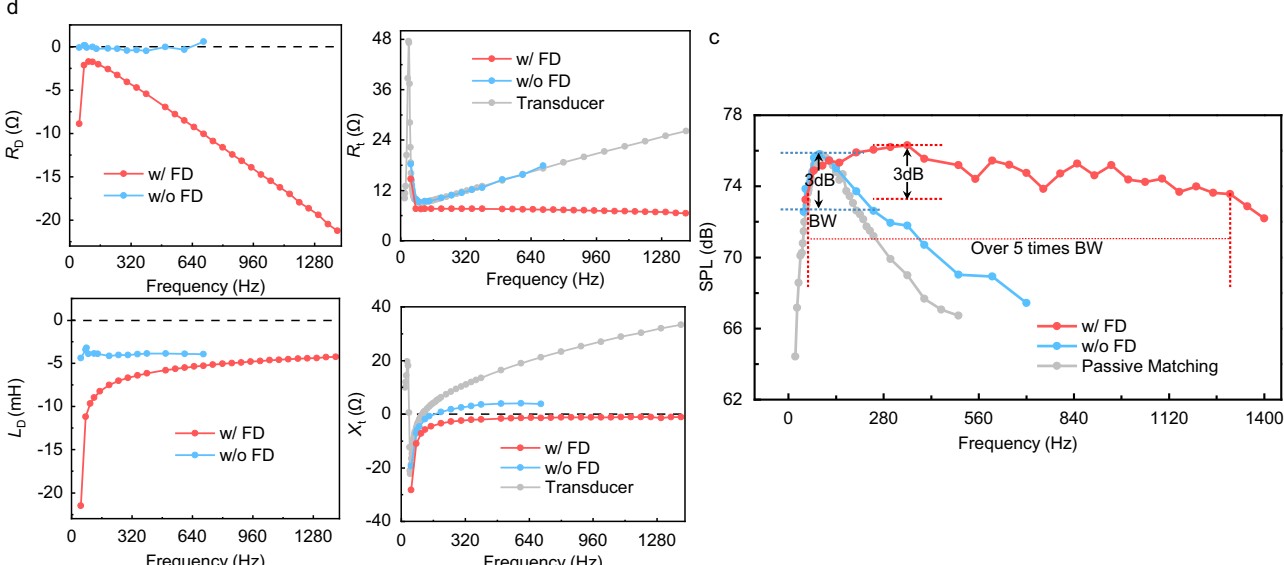

**Fig. 4 | Experimental verification for digital non-Foster-inspired electronics.** **a** Photograph of the proposed digital non-Foster-inspired circuitry, which consists of four SiC MOSFETs ($S_1$ - $S_4$), passive elements (capacitors $C_{DC}$ and $C_{AC}$, LC filter $L_s$ and $C_s$), and the signal processing setup, which includes gate drivers, a DSP, a DC/DC module, ADC, V (voltage) and I (current) sensors. **b** Photograph of the experimental setup. The laser sensor, oscilloscope and power analyzer are used for measurement and recording of experimental data. The power amplifier voltage $u_s$ is generated by a power amplifier and controlled by a signal generator. $U_{DC}$ is provided by a DC source. The rest include an auxiliary power supplier (APS), a personal computer (PC) and an electrical insulation transformer. **c** Negative resistance and inductance with or without FD synthesized by the designed digital non-Foster-inspired circuit, with their respective negative impedance characteristics characterized by the power analyzer. **d** Comparison of the sound pressure level (SPL) curves under different impedance matching approaches.

---

connected with a synthetic negative resistance and inductance with FD was significantly enhanced by bandwidth expansion, each color of the self-made Hunan University photo (four Chinese characters), represented by different frequency tones, can be received. By comparison, since the system bandwidths under synthetic negative resistance and inductance without FD and passive matching are relatively narrow, the pixels of the Hunan University photo at higher tone frequencies are lost (See Supplementary Movie 1).

## Discussion

In conclusion, we have demonstrated digital non-Foster-inspired electronics by leveraging digital control techniques and switch-mode electronics. Distinct from op-amp-based analog non-Foster circuits, the proposed circuit enhances the implementation of non-Foster impedance matching. The transition from analog to digital implementation provides an opportunity to synthesize equivalent negative R and C/L with desired FD, ensuring stability and good transient performance. Furthermore, the implementation of programmable control evades tedious manual operation, which ensures active circuits with the real-time tunability. From the hardware design standpoint, the utilization of switch-mode electronics results in a leap forward in terms of power handling. Long-range image transmission experiments have validated the superiority of the proposed digital non-Foster electronics in terms of real-time tunability and high-power handling. Breaking through the limitations in operating frequency range of 4 kHz, the proposed digital non-Foster-inspired electronics may offer new opportunities for enhancing the bandwidth, power and agility of sub-wavelength resonance-based systems, also beyond acoustics, and promote practical implementation of broadband metamaterials.

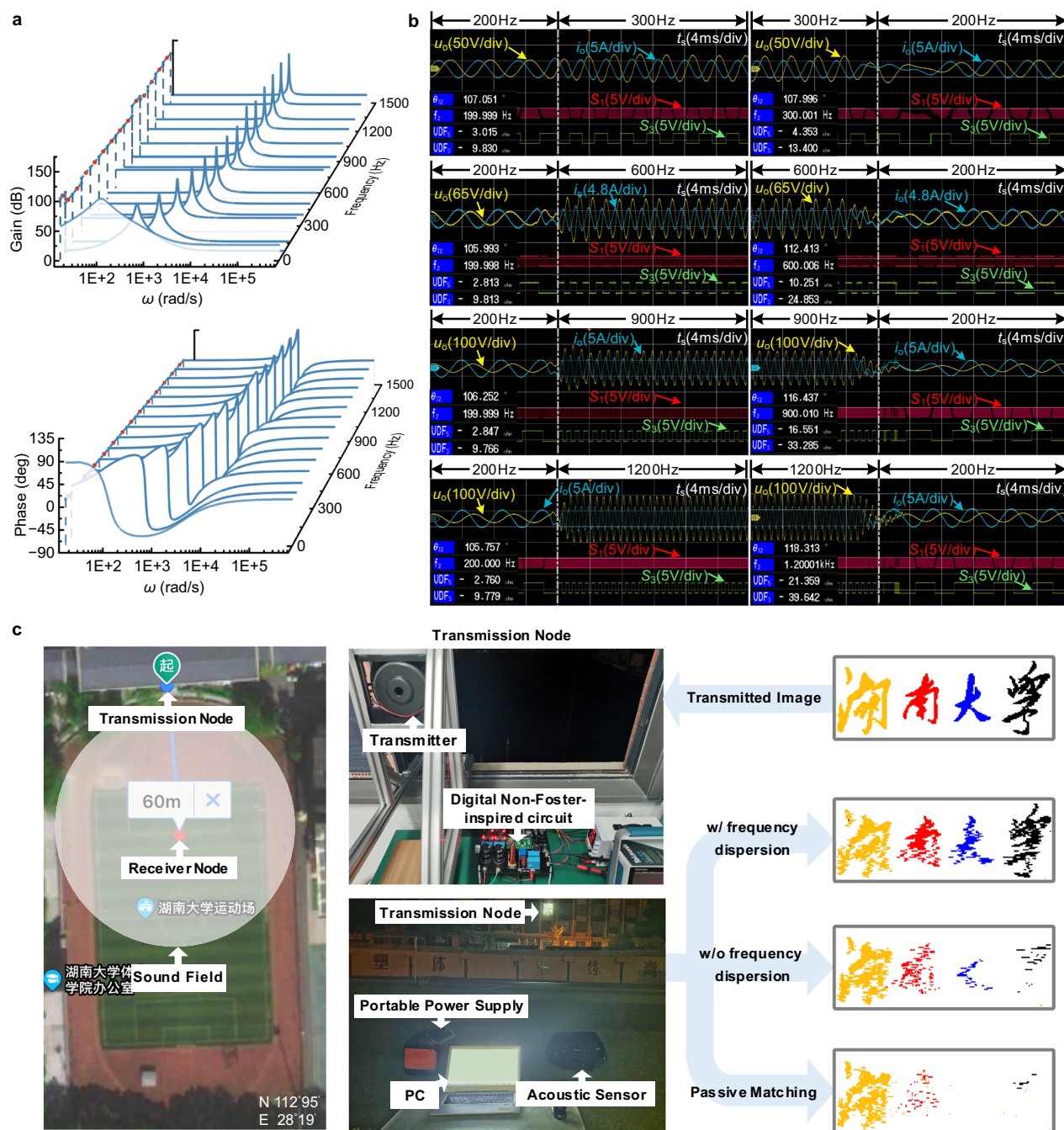

**Fig. 5 | Transient features of digital non-Foster-inspired electronics and image transmission based on high-power broadband acoustic radiation. a** Amplitude-frequency and phase-frequency curves of the self-adaptive PR controller. **b** Transient response of the digital non-Foster-inspired circuit during frequency transition. The yellow curve indicates terminal voltage $u_o$ of the digital non-Foster circuit, while the blue one is the output current $i_o$. Here, $u_o$ and $i_o$ are set to associated reference directions. The control signals of the switch devices $S_1$, $S_3$ are shown at the bottom. **c** Image transmission experiment over airborne acoustic channels. Transmitted image of the self-made Hunan University photo with a spatial resolution of $62 \times 175$ pixels. The transmitter is completed by the electroacoustic transducer fed through the designed digital non-Foster-inspired circuit, while the receiver is an acoustic sensor. The receiver node is installed to an open playground at a distance of approximately 60 meters from the transmission node, which minimized the effect of sound reflection, as well as environmental noises in reception. The transmitted and received images under different matching approaches are shown for comparison.

## Methods
### Stability analysis
To analyze the stability of loading the electroacoustic transducer with the digital non-Foster-inspired electronics, the general circuit structure shown in Fig. 3a is simplified as its transfer function block diagram in Supplementary Fig. 2a. According to Mason's gain formula, the block function diagram can be simplified as shown in Supplementary Fig. 2b. The corresponding open-loop transfer function block diagram is illustrated in Supplementary Fig. 2c. By evaluating the pole locations of the transfer function as the four key control parameters, including $K_p$, $K_r$, $K_f$, and $w_c$, varied, we can observe the parameter ranges that are capable to assure the system stable operation. The root locus can be plotted as shown in Supplementary Fig. 2d, e. The open-loop gain/phase margins can be plotted as shown in Supplementary Fig. 2f.

Consequently, the parameter design of PR controller for stability is easy to achieve as elaborated in Supplementary Note 2.

### Software design for the digital non-Foster-inspired electronics

The software design primarily consists of the main program and interrupt service routines (ISRs). A DSP (TMS320F28335, Texas Instruments) is used to execute programs here. In the main program (Supplementary Fig. 3a), initialization of peripherals and PR controller, as well as the relevant interrupt configuration, is complete. Afterwards, the DSP remains in a waiting state until interrupts occur. The purpose of the timer interrupt is to generate the PWM output pulses, and the period of the pulse is equal to the timer period. Whenever the counter overflows, an interrupt will be generated by the timer. Then, an interrupt request will be sent to the CPU where the ISR is handled, once it is idle. Meanwhile, the program pointer will be automatically located to the starting address of the ISR function. The primary work of the timer ISR, shown in Supplementary Fig. 3b. The detailed steps are elaborated in Supplementary Note 3.

### The experimental design of image transmission over airborne acoustic channels

The image transmission experiments over airborne acoustic channels are carried out with the frequency shift keying (FSK) technique[52]. The receiver node is installed to an open playground at a distance of approximately 60 meters from the transmission node, which minimized the impact of acoustic reflection and environmental noises on reception, as illustrated in Fig. 4c of the main text. The operation of the complete experiment is introduced in Supplementary Note 8.

### Reporting summary

Further information on research design is available in the Nature Portfolio Reporting Summary linked to this article.

## Data availability

Source data are provided with this paper.

## Code availability

The code to implement the DSP control of our digital non-Foster-inspired circuit as well as all the Matlab codes for the experiment of image transmission over airborne acoustic channels is available at https://doi.org/10.5281/zenodo.11120641.

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

## Acknowledgements
X.Y. would like to thank the support given by a National Natural Science Foundation of China grant (52077071) and a grant (2019RS1028) from Department of Science and Technology of Hunan Province, China.

## Author contributions
X.Y., X.Z., X.O. and A.A. conceived and developed the concept. X.Y., Z.Z. and M.X. designed and built the non-Foster-inspired circuit. X.Y. planned and directed the experiments. X.Y., Z.Z., M.X., X.Z., S.L. and Y.Z. performed the experiments. X.Y., Z.Z., and S.L. performed the simulation and theoretical analysis. X.Y., Z.Z. and M.X. wrote the initial draft of the paper. X.Y., Z.Z., Y.Z. and A.A. contributed to interpreting the results, editing, and revising the manuscript.

## Competing interests
The authors declare no competing interests.
