## [Peer Review File · Nature Communications]

REVIEWER COMMENTS

Reviewer #1 (Remarks to the Author):

The PR+FD approaches to non-Foster compensation are significant to the field and noteworthy. The experimental demonstrations validate application to high-power acoustic transmitters.

Detailed comments:

- 1) Please state that the overall PR and PR+FD method may be limited to transmitter applications, due to the need for frequency estimation to tune the PR+FD. It would seem difficult to estimate frequency at low signal-to-noise in receiver applications with multiple input frequencies present, and in situations where the desired signal may not be the strongest frequency present at a receiver input.
- 2) What are the advantages and disadvantages of the proposed adaptive PR approach relative to other prior-art digital non-Foster methods such as in doi.org/10.1109/OJCAS.2022.3174694 or doi.org/10.1109/SECON.2017.7925352
- 3) Please indicate component values on all the schematics to allow researchers to reproduce the experiment. What are L_s , C_{dc} , and C_s in Fig 3a? What are the part numbers for the switches S1-S4 in Fig 3a? What are the part numbers for V_{sensor} and I_{sensor} in Fig. 3a? What is the part number for the electroacoustic transducer? Please indicate TMS320F28335 on the DSP block of Fig 3a. What are CMS and MMD in Fig 2a?
- 4) To allow readers to compute the Chu limit, what is the size of the size of the electroacoustic transducer? What is Chu parameter " ka " at the passive-tuned frequency? What was the passive-tuned frequency (approx. 100 Hz)? From the figures, it seems that ka is fairly large, and the Chu limit is fairly broad bandwidth. Also, should the reader assume an acoustic monopole (vs dipole) Chu limit?
- 5) What is the instantaneous closed-loop bandwidth of the PR+FD?
- 6) P. 3 line 6: "device footprint" may be more clearly stated "device size"
- 7) P. 3: Please provide citations of references that validate the claims of shortcomings in power handling and stability in this sentence: "In addition, power handling and stability limitations of analog non-Foster circuits hinder the implementation of a broad range of metamaterial devices in which bandwidth is important"
- 8) P3 line 11: consider changing to "an effective negative resistance, negative inductance or negative capacitance with on-demand FD."
- 9) P3 line 21: consider "in operation" in place of "in operando"

10) P4 line 4: please more clearly state that commercially available component values and tolerances of passive analog RLC elements limit the attainable negative impedances of analog non-Foster circuits to corresponding discrete values

11) P5 line 2: “high-power scenarios” is probably best limited to “high voltage scenarios,” since large currents and low voltages can be high power in applications where impedances may be low

12) P5: please consider rewording more clearly “allow operations exempt from delicate manual tuning”

13) P5: consider rephrasing/clarifying “However, constrained by the available passive analog RLC elements, the negative impedance attainable with analog non-Foster circuits may not span a continuous range, and their tuning often depends on tedious circuit component replacements. Therefore, analog non-Foster matching networks may not synthesize arbitrary FD, facing challenges with discretized parameter space and complicated manual tuning.” It seems the authors are trying to state that RLC components are commercially available only at discrete values, therefore constraining non-Foster circuits to corresponding discrete values.

14) P7 Eq (1): the description of variable V_D does not say whether it is peak, peak-peak, rms, etc. Is it a complex phasor?

15) P7 Eq (1): units are not given for volume velocity V_D or impedance Z_{rad} or power P_{ar} . Please avoid the ambiguous impedance units of Rayl, since there are 2 different definitions in use. Please specify impedance units such as $N \cdot s/m^3$

16) Please check to add units to all defined variables throughout the paper

17) P7 Eq (2), in text below Eq 2 it would be helpful to just say $Z_D(\omega) = R_D(\omega) + LD(\omega)$ if that is true,

18) P7 bottom: it is unclear which of the two preceding sentences “this” is referring to: “This results in a reduction in radiation efficiency and considerably limits the operational bandwidth”

19) P10: u_0 and i_0 are not defined?

20) P10: where authors state “The operation frequency f_s is first acquired”, please comment on the acquisition time and the accuracy/precision of the estimated frequency. Also, would the accuracy of the frequency estimate set a lower bound on the closed-loop bandwidth and/ or on the resonator cutoff bandwidth set by ω_c ?

21) P10: if the frequency being referred to in the following sentence is the operating frequency f_s , please say so: “a large gain at an arbitrary known frequency”

22) P10: in the phrase “ has a fast dynamic performance and stability” please state some specific quantities such as closed-loop bandwidth for speed, and such as gain/phase margins for stability.

23) P11: please also state the center frequency of the passive-tuned case in “with a radiation bandwidth of 49 Hz”

24) P13: if the “transient process lasts only two or three cycles” in estimating f_s and changing the resonance filter, would this imply that the instantaneous bandwidth of the proposed adaptive PR approach is approximately $f_s/3$? If so, the reader should be cautioned that the bandwidth of the Fig 3e

curve denoted “w/ FD” is not indicating instantaneous bandwidth, whereas the curve denoted “w/o FD” does actually represent an instantaneous bandwidth

25) Supplement P6: poles instead of roles in “the system roles will turn”

Reviewer #2 (Remarks to the Author):

The comments are given in attached PDF file.

Reviewer #3 (Remarks to the Author):

In this work, digital control techniques combined with switched mode circuits are applied to implement a non-Foster reconfigurable digital impedance matching network. As a result, the self-adaptive proportional resonant controller replaces the need for manual operation with in-situ tunability and improves power handling.

In general, the developed of the paper is well conducted, as well as the results obtained. But there are some issues and observations that authors have to clarify and improve the manuscript.

a) The presentation of the captions of the Figures and their content should be expressed more clearly. As shown now they are very confusing. Please improve them.

b) Please show in a table the part number or model of the MOSFET, module driver, device used to implement DSP, current sensor, voltage sensor, inductors, capacitors, etc.

c) As a suggestion: The software program flowchart of the digital non-Foster circuit, shown in Extended Data Figure 3 can be added as very useful information to Figure 3 of the article document.

d) In special I have some observations about Extended Data Section Figure 2 (Stability analysis of the digital non-Foster electronics):

In control systems, it is typically used to represent a signal that is fed back to two different summing points as two independent loops and not in the form presented in diagrams a, b, and c, where a single loop has two outputs to the summing point. Please correct.

Some of the simplifications shown in b, c, d, and e based on block algebra are correct, but the same result could be reached in a simpler way by applying the properties of feedback in summing points to the simplification of loops. The block diagram shown in figure f also can simplified because the functions of

input branches are connected in parallel as a sum. Please express the final result of the block diagram of transfer functions using the property of linearity of the system.

(Reference of the latter two comments: Feedback Control Systems, John van De Vegte, Prentice Hall, Third Edition, Chapter 3)

NCOMMS-23-41169 Xin Yang, Zhihe Zhang, Mengwei Xu, Shuxun Li, Yuanhong Zhang, Xue-Feng Zhu², Xiaoping Ouyang, and Andrea Alù : **'Digital Non-Foster Electronics for Broadband Impedance Matching'**

This manuscript presents an interesting and certainly very useful method for increasing the radiated power of an originally mismatched radiator over a wide bandwidth. The results presented are correct and convincing and can be very important for practical engineering systems. There is also no doubt that the presented approach is far more flexible than the widely studied linear non-Foster matching and that it offers higher operating power levels and better efficiency. Although the manuscript is based on some well-known ideas, I believe that the presented research results are novel enough to be published in Nature Communications.

In the present form of the manuscript, however, there are several serious terminological problems that may lead the reader to wrong conclusions. In addition, there are some previous representative studies with similar ideas that, at least in my opinion, should be included in the reference list. Finally, some important limitations of the presented approach are not emphasized. In light of all this, I recommend a comprehensive revision of this manuscript.

My specific comments are given below:

1) Terminology problems

- a) I think that the proposed title ("Digital Non-Foster Electronics for Broadband Impedance Matching") is inappropriate. Foster's theorem states that the reactance and susceptance of a lossless network always increase with frequency. Of course, this theorem only applies to networks without resistors. Foster's theorem does not apply to lossy networks (the phenomenon of anomalous dispersion is a very well-known example). In other words, the Foster networks can only contain ordinary (positive) capacitors or/and inductors (or, in the distributed element scenario, the sections of lossless transmission lines). On this basis, the term non-Foster elements is used in the literature almost exclusively for electronic circuits that behave like negative capacitors and negative inductors, but (in an ideal case) not like negative resistors. There are some authors who place negative resistors within the scope of non-Foster elements, but I think this is fundamentally wrong and causes unnecessary confusion. In this manuscript, the authors make extensive use of negative resistance, which means that in addition to reactance cancellation with negative inductance and negative capacitance, the system also has the characteristics of a negative-resistance reflection amplifier. The authors say that the generated negative resistance cancels the positive resistance, which physically means that it compensates the losses with the reflection gain based on negative resistance. Therefore, I firmly believe that using the term "non-Foster matching" in this context is incorrect. The authors may use "non-Foster-like matching" or "non-Foster-inspired matching" (the terminology used in some related studies [R1, R2] that use both non-Foster elements and negative resistances for amplification) or "generalized active broadband matching with arbitrary dispersion cancellation" or something similar, but not non-Foster matching. The authors should not only change the terminology related to "non-Foster", but also make corresponding changes throughout the manuscript (e.g., clearly explain that the use of a negative resistance causes reflection amplification that boosts radiated power).
- b) The term "digital non-foster" also seems confusing, as there are several published reports [R3, R4] that use this term for systems that in fact mimic ordinary non-foster systems with linear amplifiers with positive feedback but use a digital approach. In ordinary ("analog") non-Foster systems, the negative impedance is achieved by a superposition of the original signal and 'assisting signal' fed from the output of an amplifier via positive feedback. The "digital systems" from [R3, R4] first apply an A/D conversion, process the signal in the digital domain, apply a D/A conversion and again use a superposition. So, these systems really "imitate" classical "analog" non-Foster systems. In this manuscript, however, the authors actually use high-efficiency nonlinear switching systems that provide an assisting signal by low-pass filtering a discrete signal generated in an H-bridge with MOSFET switches. This is fundamentally different from the studies in the literature, but is similar to the patent in [R5]. Therefore, I believe that the term "switching mode" or "digitally controlled switching mode" or similar is much more appropriate than simply "digital". This would clearly differentiate this work from other studies in the literature.

2) Referencing problems

It seems that the authors mainly used references from the physics community. However, I believe that there are several very relevant papers from the engineering field that use similar ideas. For example, the concept of using additional sources that simultaneously implement non-Foster cancelation and negative resistance amplification (which is actually a linear, less efficient version of the authors' approach) was introduced in [R2]. In addition, an idea for a purely "digital" approach was presented in [R4]. A non-Foster approach with switching mode and H-bridge was presented in [R5]. There are several other examples that can be found in the IEEEXplore database. In my opinion, the authors should pick up a few relevant engineering papers with similar ideas and include them in the reference list. I would like to be perfectly clear: it would simply be very good to include some previous studies with similar ideas, just to put the authors' work in context. I have gone through all these studies carefully and none of them went as far with experimental investigation and practical implementation. Therefore, I strongly believe that the authors' work is original enough to be published in Nature Communications (after implementing the suggested changes)

3) Lack of discussion on limitations of Authors' approach

The core idea of the authors' approach, which allows almost arbitrary frequency dispersion of the "synthesized" negative capacitance, inductance and resistance, is "on-the-fly" control by a digital PI regulator. Obviously, the highest operating frequency depends on the speed of a DSP control loop. The authors have clearly shown that this approach works in the audio range (up to 2 kHz). Is this the highest possible frequency? There are commercial digital shortwave transmitters with power amplifier/modulator that use MOSFET-based H-bridges up to 30 MHz. Would it be possible to apply the author's approach to these or even higher frequencies? I think it would be good to include a couple of sentences discussing the limitations imposed by available DSP technology.

REFERENCES

- [R1] A. Kirichenko and S. Hrabar, "Non-Foster-inspired Two-transmitter System based on Coherent Current/voltage Sources," 2022 IEEE International Symposium on Antennas and Propagation and USNC-URSI Radio Science Meeting (AP-S/URSI), Denver, CO, USA, 2022, pp. 1308-1309, doi: 10.1109/AP-S/USNC-URSI47032.2022.9886333
- [R2] S. Hrabar, I. Cavlek, D. Mikulic, S. Milic and E. Sopp, "Extension of Non-Foster-inspired Two-transmitter Matching to Arbitrary Antenna Impedance," 2021 International Symposium ELMAR, Zadar, Croatia, 2021, pp. 49-52, doi: 10.1109/ELMAR52657.2021.9550906.
- [R3] T. P. Weldon, J. M. C. Covington, K. L. Smith and R. S. Adams, "Stability conditions for a digital discrete-time non-Foster circuit element," 2015 IEEE International Symposium on Antennas and Propagation & USNC/URSI National Radio Science Meeting, Vancouver, BC, Canada, 2015, pp. 71-72, doi: 10.1109/APS.2015.7304421.
- [R4] D. M. Johnson and T. P. Weldon, "A Clock-Tuned Discrete-Time Negative Capacitor Implemented Using Analog Samplers," 2018 IEEE International Symposium on Circuits and Systems (ISCAS), Florence, Italy, 2018, pp. 1-5, doi: 10.1109/ISCAS.2018.8351121.
- [R5] White et al 'Switched Mode Negative Inductor', US patent 9923548, May 2018

Response to The Comments

Manuscript ID: NCOMMS-23-41169

Title: Digital Non-Foster Electronics for Broadband Impedance Matching

Dear Reviewers,

We greatly appreciate your work on our manuscript. All the comments we received **are valuable and helpful** for improving our manuscript and enlightening our future research. We have carefully studied all the comments, **further complemented our work**, and **made major corrections**.

We have marked the corrections **in red** in the revised manuscript. Deletions in the text are not shown. In addition, we have also provided a point-by-point response to the reviewers' comments as required.

Reviewer #1:

Comments to the Author: The PR+FD approaches to non-Foster compensation are significant to the field and noteworthy. The experimental demonstrations validate application to high-power acoustic transmitters.

Detailed comments:

Response: We are sincerely grateful for the reviewer's support and guidance! We do treasure this opportunity to fully amend and polish our paper. Many thanks in advance!

Comment 1-1: Please state that the overall PR and PR+FD method may be limited to transmitter applications, due to the need for frequency estimation to tune the PR+FD. It would seem difficult to estimate frequency at low signal-to-noise in receiver applications with multiple input frequencies present, and in situations where the desired signal may not be the strongest frequency present at a receiver input.

Response 1-1: Many thanks for your suggestion! We agree our proposed approach based on PR +FD is suitable for transmitter applications, but may encounter limitations in receiver applications due to the need for frequency estimation using the voltage comparator and eCAP module of DSP 28335. As you mentioned if strong noise or interference is added at the receiver, the voltage comparator might malfunction due to input distortion. For the revision, we have emphasized the proposed approach for transmitter applications.

[1] Xiong, F. Digital Modulation Techniques, 2nd ed (Boston, MA, USA: Artech House, Inc., 2006).

Comment 1-2: What are the advantages and disadvantages of the proposed adaptive PR approach relative to other prior-art digital non-Foster methods such as in doi.org/10.1109/OJCAS.2022.3174694 or doi.org/10.1109/SECON.2017.7925352.

Response 1-2: We do apologize for the insufficient literature review and unclarity of the advantages/disadvantages of our proposed approach compared with prior-art digital non-Foster methods. Taking this chance of manuscript revision, we have added suggested references and detailed comparisons.

1) Prior-art digital non-Foster methods

Indeed, the pioneering digital non-Foster circuit element was implemented by Dr Weldon (Ref. [1]) as shown in Fig. R1. **Open-loop control method** was used in this digital approach. The analog voltage at the input terminals $v(t)$ is first digitized with an ADC (analog-to-digital converter) into discrete-time signal $v[n]$. The behavior of this circuit is then established through calculating the appropriate current using a discrete-time filter $H(z)$ within the digital signal processing. Finally, the current at the input terminals $i(t)$ is converted from discrete-time current $i[n]$ by a current-output DAC (digital-to-analog converter).

Fig. R1. Block diagram of digital discrete-time non-Foster circuit.

Furthermore, Ref. [2] analyzed the stability of the digital non-Foster circuit elements by judging the relative positions of poles of the corresponding discrete-time transfer function. Then, Ref. [3] and [4] discussed the possibility of digital series negative RC (resistor-capacitor) and digital series negative RL (resistor-inductor) circuit implementations by digital non-Foster circuit elements to improve stability or to mitigate parasitic resistance. Ref. [5] improves the accuracy and stability of the earlier digital non-Foster methods by refining the modeling of the digital non-Foster circuit elements and increasing the complexity of discrete-time filter $H(z)$. In Ref. [6], authors simulated the impedance matching using a non-Foster circuit element in series with the electrically-short monopole antenna model.

2) Differences between our proposed approach and prior-art methods

By comparison with the **open-loop control method** in prior-art digital non-Foster circuits, we believe that our proposed **closed-loop switch-mode method** has two advantages.

Firstly, our output negative capacitor or negative inductor or negative resistor is set by a reference signal, which will hardly influence the controller setting (the self-adaptive PR

approach is programmed into the digital processor).

For the prior-art digital non-Foster methods, their output negative impedance is **also determined by the parameters of the controller setting (digital signal processing)**. In Ref. [1] and [2], the signal processing block $H(z)$ must be equal to Eq. (R1) so that the output characteristic of the digital non-Foster circuit element achieves the desired negative capacitor

$$H_C(z) = \frac{C(1-z^{-1})}{T} \quad \text{R1}$$

In Ref. [3] and [4], the signal processing block $H(z)$ must be equal to Eq. (R2) or Eq. (R3) so that the output characteristic of the digital non-Foster circuit element achieves the desired digital series negative RC (resistor-capacitor) or digital series negative RL (resistor-inductor).

$$H_{RC}(z) = \frac{(R_{\text{ser}}C - R_{\text{dac}}C + T)z + (R_{\text{dac}}C - R_{\text{ser}}C)}{(R_{\text{ser}}C + T)z - R_{\text{ser}}C} \quad \text{R2}$$

$$H_{RL}(z) = \frac{(L + R_{\text{ser}}T - R_{\text{dac}}T)z - L}{(L + R_{\text{ser}}T)z - L} \quad \text{R3}$$

As the parameters of digital signal processing $H(z)$ must ensure the stability of digital non-Foster circuit element, the range of output negative impedance will be significantly limited [2].

Secondly, our proposed self-adaptive PR approach does not rely on the modeling of elements in the digital non-Foster circuit. The accuracy of the output negative capacitors, negative inductors, or negative resistors in the prior-art digital non-Foster techniques can be greatly impacted by the modeling quality and completeness of all circuit parts. In Refs. [1]-[4], the input impedance of the ADC and the output impedance of the DAC are assumed to be infinite, a sampler switch and a zero-order hold are introduced to simulate the conversion between digital and analog signals, and a time delay τ is added to the DAC output to consider the effect of the latency time delay. In Ref. [5], the modeling of the digital non-Foster circuit elements is further refined. The input impedance of the ADC and the output impedance of the DAC are represented by R_{adc} and R_{dac} respectively, which is considered not to be infinite at high frequencies above 20 MHz. For these open-loop control approaches, practical realization must take into account the purely technological issues (parasitics of the circuit layout and the attached devices, etc.), which will cause unexpected accuracy and stability issues.

On the contrary, the **closed-loop negative feedback control method** is used in our proposed approach. If the output negative impedance of our proposed circuit deviates from the desired value, it can automatically reduce or eliminate the deviation by the PR control, and ultimately realize the desired negative impedance. Our proposed adaptive PR controller has a very large gain at the operating frequency (the operating frequency is guaranteed to be equal to the resonant frequency of the controller), which ensures the zero steady-state error

tracking of the sinusoidal reference signal.

Admittedly, our proposed approach also suffers from limitations in terms of dynamic performance and applicability to higher frequencies compared to the prior-art digital non-Foster circuits. As shown in Fig. R2, the transient response process of the prior-art digital non-Foster method in Ref. [2] only requires half a cycle for the output signal calculation to reach the steady state. And it can work at higher frequency. On the contrary, the transient process of our proposed approach including the process of operating frequency estimation and dynamic response takes two to three cycles, as shown in Fig. R3.

Although it can be further optimized to approximately within one cycle by improvement of the control algorithm as mentioned in **Response 1-24**, it can only be used for **low-frequency scenarios and cannot fit it the high-frequency application, details of which is shown in Response 2-4**.

Fig. R2. The transient response process in Ref. [2].

Fig. R3. The transient response of the digital non-Foster circuit during frequency transition in the manuscript.

For the revision, please refer to the **red words** on Introduction on page 4, Line 6-12 of the revised manuscript and on Supplementary Section 1 on page 3, Line 4 of the revised Supplementary Information.

- [1] Weldon, T. P., Covington, J. M., Smith, K. L. & Adams, R. S. Performance of digital discrete-time implementations of non-Foster circuit elements. *In 2015 IEEE International Symposium on Circuits and Systems (ISCAS)*, 2169-2172 (2015).
- [2] Weldon, T. P., Covington, J. M., Smith, K. L. & Adams, R. S. Stability conditions for a digital discrete-time non-Foster circuit element. *In 2015 IEEE Antennas and Propagation Society International Symposium (APSURSI)*, 71-72 (2015).
- [3] Kehoe, P. J., Steer, K. K. & Weldon, T. P. Thevenin forms of digital discrete-time non-Foster RC and RL circuits. *In 2016 IEEE Antennas and Propagation Society International Symposium (APSURSI)*, 191-192 (2016).
- [4] Kehoe, P. J., Steer, K. K. & Weldon, T. P. Stability analysis and measurement of RC and RL digital non-Foster circuits with latency. *In SoutheastCon 2017*, 1-4 (2017).
- [5] Daniel, C. G. & Weldon, T. P. A stable digital impedance circuit design method for resistive source impedances. *IEEE Open J. Circuits Syst.* **3**, 109-114 (2022).
- [6] Steer, K. K., Kehoe, P. J. & Weldon, T. P. Investigation of an adaptively-tuned digital non-Foster approach for impedance matching of electrically-small antennas. *In SoutheastCon 2017*, 1-5 (2017).

Comment 1-3: Please indicate component values on all the schematics to allow researchers to reproduce the experiment. What are L_s , C_{dc} , and C_s in Fig 3a? What are the part numbers for the switches S1-S4 in Fig 3a? What are the part numbers for V_{sensor} and I_{sensor} in Fig. 3a? What is the part number for the electroacoustic transducer? Please indicate TMS320F28335 on the DSP block of Fig 3a. What are C_{MS} and M_{MD} in Fig 2a?

Response 1-3: Sorry for the confusion. As the DC-link capacitor, C_{DC} , which is used to smooth the voltage ripple of V_{DC} , consists of two 450 V 820 μ F electrolytic capacitors connected in parallel with two 350 V 10 μ F film capacitors. The inductor L_s (equal to 2 mH) and the capacitor C_s (equal to 2 μ F) together form a LC filter to suppress higher-order harmonics of u_{AB} . Furthermore, the part numbers for the SiC MOSFET switches S1-S4 is C3M0060065D, Cree Inc. The voltage sensor and current sensor are TV16E (produced by Dechang Electric Co., Ltd, China) and LA35-NP (produced by LEM Electronics, Switzerland) respectively. Also, the part number of electroacoustic transducer used in the manuscript is DL100LLB-01, produced by Dongguan Huachuang Audio Equipment Co., Ltd, China.

C_{MS} and M_{MD} represents the mechanical compliance of the diaphragm suspension and the mechanical mass of the diaphragm and voice-coil assembly, respectively.

Fig. R1. Photograph of the proposed digital non-Foster circuitry.

Based on reviewer's comments, the part numbers or values of the main components as shown in Fig. R1 and electroacoustic transducer have been listed in the table R1. And the table has been added in the revised manuscript.

Table R1 The part numbers or values of the main components

Component	Part number or value	Producer
C_{AC}	350 V 10 μ F film capacitors	TDK Electronics AG, Germany
Current sensor I sensor	LA35-NP	LEM Electronics, Switzerland
Voltage sensor V sensor	TV16E	Dechang Electric Co., Ltd., China
DC/DC module	VRA2415ZP-10WR3	MORNSUN Technology Co., Ltd., China
	IB2403LS-1WR3	MORNSUN Technology Co., Ltd., China
ADC module	AD7656	Analog Devices, Inc., America
DSP	TMS320F28335	Texas Instruments, Inc., America
Module driver	UCC21520	Texas Instruments, Inc., America
SiC MOSFET	C3M0060065D	Wolfspeed, Inc., America
DC-link capacitor C_{DC}	450 V 820 μ F electrolytic capacitors	Nippon Chemi-Con Corp., Japan
	350 V 10 μ F film capacitors	TDK Electronics AG, Germany
Filter inductors L_s	2 mH	-
Filter capacitors C_s	630 V, 2 μ F film capacitor	KNSCHA Electronics Co., Ltd., China
Electroacoustic transducer	DL100LLB-01	Dongguan Huachuang Audio Equipment Co., Ltd, China

Many thanks for your kind help! The relevant content has been added in the revised Supplementary Information. Please refer to the **red words** on Extended Data Table 3 on page 29 of the revised Supplementary Information.

Comment 1-4: To allow readers to compute the Chu limit, what is the size of the size of the electroacoustic transducer? What is Chu parameter “ka” at the passive-tuned frequency? What was the passive-tuned frequency (approx. 100 Hz)? From the figures, it seems that ka is fairly large, and the Chu limit is fairly broad bandwidth. Also, should the reader assume an acoustic monopole (vs dipole) Chu limit?

Response 1-4: We are grateful for the constructive suggestion!

1) First of all, we apologize for the mistake of the description of Fig. R1 (corresponding to Fig. 3e in the original manuscript) about the passive-tuned experimental procedure in Page 11 Line 19 in the original manuscript. The description “A total capacitance of 1000 μ F was used to impedance match, with a radiation bandwidth of 49 Hz.” should be corrected to “Here, the total capacitance value is equal to 1000 μ F, which is intended to compensate the eddy current inductance of the acoustic radiator at 49Hz.” 49 Hz is the resonant frequency of the acoustic radiator.

Fig. R1 Comparison of the sound pressure level (SPL) curves under different impedance matching approaches and the measured vibration mode of the transducer using a laser sensor (inset figure).

2) Passive-tuned matching experiment

The experiment considering the electroacoustic transducer matched with capacitors has been carried out, i.e. the passive-tuned experiment. We aim to reduce the eddy current impedance effect on the radiation efficiency and operating bandwidth by compensating the eddy current inductance $L_E(\omega)$ of the acoustic radiator at 49Hz. When $f_s=49$ Hz and $L_E=9.851$ mH, the capacitance for passive-tuned matching can be calculated by Eq. (R1)

$$C_p = \frac{1}{\omega^2 L_E} = \frac{1}{(2 \cdot \pi \cdot 49)^2 \cdot 0.009851} = 1070.95 \mu\text{F}$$

R1

In the experiment, the total capacitance value can only be set to $1000 \mu\text{F}$, which is realized by using ten $100 \mu\text{F}$ 1210 chip capacitors in parallel as shown in Fig. R2. Please note $1070.95 \mu\text{F}$ is difficult to set perfectly due to the limited availability of commercial capacitors (also the shortage of analogue non-foster circuitry). So the actual resonant frequency turns to 50.98 Hz .

Then, thus vibration velocities of the transducer passively-compensated were measured with the same excitation voltage. SPL (Sound Pressure Level) was calculated with the measured transducer velocities as described in **Supplementary Section 4 (Conversion of vibration velocity to sound pressure)**, and the result is displayed in Fig. R1.

Fig. R2 The conventional passive-tuned matching capacitor

3) Derivation of the acoustic Chu limit

Ref. [1] provides a derivation of the acoustic Chu limit. According to [2-3], the electroacoustic transducer can be treated as an acoustic monopole model. Following the derivation of the acoustic Chu limit, the quality factor for multipolar orders 0 to 3 as shown in Fig. R3. As the total Q is a weighted sum of these individual values, as a function of the different contributions of each harmonic to the overall radiation, the minimum achievable Q is Q_0 . Therefore, the minimum quality factor for a small acoustic radiator scales with the radius of the sphere enclosing the source as $1/ka$, where k is the wavenumber and a is the radius, and it can be approached for a purely monopolar source. In order to approach this lower bound, a passive acoustic source must store as little energy as possible in the region $r < a$. As the size of a passive radiator shrinks, the fields at resonance grow close to the origin, contributing more to the near-field stored energy than to the radiated energy. This fundamentally limits the overall achievable bandwidth.

	Electromagnetic Q	Acoustic Q
	-	$\frac{1}{ka}$
	$\frac{1}{ka} + \frac{1}{(ka)^3}$	$\frac{2}{ka} + \frac{2}{(ka)^3}$
	$\frac{3}{ka} + \frac{6}{(ka)^3} + \frac{18}{(ka)^5}$	$\frac{4}{ka} + \frac{9}{(ka)^3} + \frac{27}{(ka)^5}$
	$\frac{6}{ka} + \frac{21}{(ka)^3} + \frac{135}{(ka)^5} + \frac{675}{(ka)^7}$	$\frac{7}{ka} + \frac{27}{(ka)^3} + \frac{180}{(ka)^5} + \frac{900}{(ka)^7}$
Chu bound	 $\frac{1}{(ka)^2}$	 $\frac{1}{ka}$

(a)

(b)

Fig. R3 Quality factors for multipolar orders 0 to 3. (A) Expressions for acoustic and electromagnetic multipoles. The Chu bound (final row) is the lowest possible Q in the limit $ka \ll 1$. (B) Plot of the acoustic Q showing the divergence for subwavelength radiators. The lowest curve, corresponding to Q_0 , is the ultimate bound for acoustic radiators.

3) Calculation of the acoustic Chu limit in the passive-tuned matching experiment

The part number of electroacoustic transducer used in the manuscript is DL100LLB-01, produced by Dongguan Huachuang Audio Equipment Co., Ltd. with a radius of the radiant surface of 0.103m. Based on the above analysis, Chu limit of the acoustic monopole transducer at 50.98 Hz can be approximately calculated by Eq. (R2) with ka (wavenumber multiplied by radius)

$$Q \geq \frac{1}{ka} = \frac{342}{2 * \pi * 50.98 * 0.103} = 10.37 \quad \text{R2}$$

ka is around 0.092 so the bandwidth is comparatively narrow. For the revision, please refer to the **red words** on Section “Experimental verification of digital non-Foster-inspired electronics” on page 14, Line 20-23 of the revised manuscript.

- [1] Rasmussen, C. & Alù, A. Non-Foster acoustic radiation from an active piezoelectric transducer. *Proc. Natl. Acad. Sci. USA*. **118**, e2024984118 (2021).
- [2] Harris, N. J. & Hawksford, M. O. J. Introduction to distributed mode loudspeakers (DML) with first-order behavioural modelling. *IEE Proc.-Circuits Devices Syst.* **147**, 153-157 (2000)
- [3] Leach, W. M. Introduction to electroacoustics and audio amplifier design 85-105 (Kendall Hunt, 2010).

Comment 1-5: What is the instantaneous closed-loop bandwidth of the PR+FD?

Response 1-5: We are sorry for the unclear statement on the controller bandwidth of self-adaptive proportional-resonant (PR) approach and the control gain curve with operating frequency f_s .

1) Gain curve by self-adaptive PR controller

In order to ensure that our approach can accurately output the desired negative impedance over a wide frequency range, the controller is requested to provide infinite gain to eliminate the sinusoidal steady-state error. However, the conventional PR controller can a considerable gain only at a resonant frequency, but the gain drops sharply when the operating frequency of the electroacoustic transducer deviates from the resonance frequency.

In digital modulation technique transmitting applications, the modulated signals emitted by transmitters are composed of single-frequency signals with different amplitudes, different phases, and different frequencies in series at a given time [1]. Based on this characteristic, we propose the self-adaptive PR controller. Our proposed self-adaptive PR controller can not only exhibit the conventional PR controller characteristics at each instant time, but it can automatically adjust the resonance frequency according to the modulated signals so as to realize high gain over a wide spectrum.

The conventional PR controller with the cutoff bandwidth ω_c around the resonant frequency can be expressed in the s-domain as:

$$G_{PR}(s) = K_p + K_r \frac{2\omega_c s}{s^2 + 2\omega_c s + \omega_0^2}$$

R1

where K_p and K_r are the proportional term and the integral term of the controller, respectively. $\omega_0 = 2\pi f_0$ is the fundamental angular frequency.

Assuming that the frequency of the modulated signal is f_{s1} at t_0 , the frequency characteristics of the PR controller at this time are shown as the red line in Fig. R1. Through frequency estimation and controller parameter adjustment, the resonant frequency f_0 of PR controller is equal to the modulated signal frequency f_{s1} at this moment, i.e., $f_0 = f_{s1}$, and thus the gain of PR controller at the modulating signal frequency is G_1 and the phase is zero. The modulated signal frequency is switched from f_{s1} to f_{s2} at t_1 , and then the PR controller automatically adjusts the frequency characteristics as shown in the blue line of Fig. R2. After frequency switching, the resonant frequency of the PR controller is still kept equal to the modulated signal frequency by adjusting controller parameters, i.e., $f_0 = f_{s2}$, and then the gain of PR controller at the modulating signal frequency is G_2 and the phase is zero.

Fig. R1 Amplitude-phase response curves of the adaptive PR controller before and after frequency

switching.

Fig. R2 (corresponding to Fig. 4a and b in the original manuscript) shows that the operating gain of the controller is theoretically kept as the peak gain at the resonant frequency, and the phase is kept as zero with real-time adaptive adjustment of the PR controller according to the frequency of the modulated signal (i.e., the operating frequency of the electroacoustic transducer). Thus, the gain-phase curves with the operating frequency f_s for self-adaptive PR approach and frequency dispersion can be obtained as shown in Fig. R3. Our method does exhibit a high gain over a wide spectrum since the controller parameters vary in real time according to frequency.

Fig. R2 Bode plots of the self-adaptive PR controller and the operating gain-phase curves with operating frequency f_s .

Fig. R3 Operating gain and phase curves with the operating frequency f_s .

2) Close-loop bandwidth

The close-loop bandwidth can be analysis on linearized time-invariant system [2]. In other words, the time-varying systems cannot be analyzed. Based on the closed-loop transfer function shown in Eq. (R2) and the parameter configurations in Table R1 (details as shown in **Response 1-22**), the closed-loop bode plot of the control system at an operating frequency of 900 Hz can be obtained as shown in Fig. R4. For a bode plot without DC gain, the closed-loop bandwidth is defined as the frequency range from peak gain to -3 dB. Therefore, it can be seen from Fig. R4 that the closed-loop bandwidth of the control system in 900 Hz is approximately equal to 20π rad/s. This agrees with the characteristics of the conventional PR and does not exhibit an instantaneous close-loop bandwidth.

Fig. R4. The closed-loop bode plot of the digital non-Foster electronics in the operating frequency of 900 Hz.

The relevant content has been added in the text in the revised Supplementary Information. Please refer to the **red words** on Supplementary Section 2 on page 6, Line 11-14 of the revised Supplementary Information.

- [1] Xiong, F. Digital Modulation Techniques, 2nd ed (Boston, MA, USA: Artech House, Inc., 2006).
 [2] Nise, Norman S. Control systems engineering, 8th ed. (New York: John Wiley & Sons, 2020).

Comment 1-6: P. 3 line 6: “device footprint” may be more clearly stated “device size”.

Response 1-6: We agree with the reviewer and accept the comments. The word “**device footprint**” has been changed into “**device size**” in the revised manuscript. Specifically, we have changed the sentence to “Similarly, a general constraint between the operational bandwidth and the device size exists in acoustic and microwave metamaterials.”

For the revision, please refer to the **red words** on Introduction on page 3, Line 6 of the revised manuscript.

Comment 1-7: P3: Please provide citations of references that validate the claims of shortcomings in power handling and stability in this sentence: “In addition, power handling and stability limitations of analog non-Foster circuits hinder the implementation of a broad range of metamaterial devices in which bandwidth is important”.

Response 1-7: As suggested, we have now added the appropriate references to support the statements made on Page 3.

In Ref. [1], authors state that analog non-Foster circuits are active circuits with nonlinear transistors which can introduce undesirable effects at high signal power levels. The effects of nonlinearity of non-Foster circuit have been identified in terms of match degradation, gain degradation, and potential instability. It has been observed that the non-Foster circuit impedance changes considerably as the signal power increases, thus reducing its matching capability as well as introducing stability problems.

In Ref. [2] authors indicate that the analog non-Foster elements are realized based on some kind of an amplifier with positive feedback, which is prone to instability. In Ref. [3], authors point out that the practical realization of analog non-Foster circuit must take into account the purely technological issues (parasitics of the circuit layout and those of the attached devices, etc.), which will cause unexpected instability issues.

The added references are as follows:

- [1] Jacob, M. M. & Sievenpiper, D. F. Non-Foster matched antennas for high-power applications. *IEEE Trans. Antennas Propag.* **65**, 4461-4469 (2017).
- [2] Hrabar, S., Krois, I. & Zanic, D. Improving stability of negative capacitors for use in active metamaterials and antennas. In *2018 IEEE International Symposium on Antennas and Propagation and USNC-URSI Radio Science Meeting (AP-S/URSI)*, 1901-1902 (2018).
- [3] Ugarte-Munoz, E., Hrabar, S., Segovia-Vargas, D. & Kirichenko, A. Stability of non-Foster reactive elements for use in active metamaterials and antennas. *IEEE Trans. Antennas Propag.* **60**, 3490-3494 (2012).

For the revision, please refer to the **red words** on Introduction on page 3, Line 22-25 of the revised manuscript.

Comment 1-8: P3 line 11: consider changing to “an effective negative resistance, negative inductance or negative capacitance with on-demand FD”.

Response 1-8: We agree with the reviewer and accept the comments. The word “**inductance or capacitance**” has been changed into “**negative inductance or negative capacitance**” in the revised manuscript.

Many thanks for your kind help! For the revision, please refer to the **red words** on Introduction on page 4, Line 16 of the revised manuscript.

Comment 1-9: P3 line 21: consider “in operation” in place of “in operando”.

Response 1-9: We agree with the reviewer and accept the comments. The word “**operando**” has been changed into “**operation**” in the revised manuscript.

For the revision, please refer to the **red words** on Introduction on page 3, Line 21 of the revised manuscript.

Comment 1-10: P4 line 4: please more clearly state that commercially available component values and tolerances of passive analog RLC elements limit the attainable negative impedances of analog non-Foster circuits to corresponding discrete values.

Response 1-10: Many thanks for the inspiring suggestions!

Fig. R1 shows the block diagram of an ideal negative impedance converter (NIC).

Fig. R1 the block diagram of an ideal negative impedance converter.

The impedance Z_L at port two is converted by NIC so:

$$Z_{in} = -KZ_L$$

R1

where $K > 0$ is the impedance conversion factor.

According to Eq. (R1), the port impedance Z_{in} of the NIC is dependent on the impedance conversion factor K and Z_L . For an ideal analog non-Foster circuit, $-K$ can be realized by the special structure of operational amplifier or transistor design. Z_L is composed of passive analog RLC elements, which have specific specifications and tolerance accuracy. For example, capacitance specifications for common passive capacitor packages include 100pF, 150pF, 220pF, 330pF, 470pF, 680pF, 1nF, 1.5nF, 2.2nF, 3.3nF, 4.7nF, 6.8nF, 1μF, etc. with the tolerances of $\pm 1\%$, $\pm 5\%$, $\pm 10\%$, or $\pm 20\%$, etc [1]. Similarly, passive resistors and inductors are also available in specific packages and specifications. In order to obtain different impedance values from the specific specifications, different passive analog RLC elements have to be connected in series and parallel, which result in greater errors.

As the passive analog RLC elements have values with specific specifications, analog non-Foster circuits are limited in their discretized parameter space and hence are unable to engineer accurate and arbitrary frequency dispersion. They can only be configured in a specific and constrained manner due to the limited choice of available circuit components as shown in Fig. R2.

Fig. R2. Diagram of available negative impedances by varying specific analog components.

In order to avoid possible confusion, we have changed the related sentence, please refer to the **red words** on Section “Digital non-Foster-inspired electronics” on page 6, Line 7-9 of the revised manuscript.

[1] AVX. Datasheet of Surface Mount Ceramic Capacitor Products. <https://www.kyocera-avx.com/resources/catalogs/>.

Comment 1-11: P5 line 2: “high-power scenarios” is probably best limited to “high voltage scenarios,” since large currents and low voltages can be high power in applications where impedances may be low.

Response 1-11: We agree with the reviewer and accept the comments. The word “high-power scenarios” has been changed into “**high-voltage scenarios**” in the revised manuscript to correspond to Fig. R1 (Corresponding to Fig. 1a in the original manuscript).

In order to avoid possible confusion, we have changed the related sentence, please refer to the **red words** on Section “Digital non-Foster-inspired electronics” on page 6, Line 3 of the revised manuscript.

Comment 1-12: P5: please consider rewording more clearly “allow operations exempt from delicate manual tuning”.

Response 1-12: Sorry for the unclear expression. “delicate manual tuning” was meant to change the circuit component (like resistor, capacitance or inductance) or tune variable capacitor to change the negative impedance set by analogue non-foster circuit.

In order to avoid possible confusion, we have changed the related sentence, please refer to the **red words** on Section “Digital non-Foster-inspired electronics” on page 6, Line 16-17 of the revised manuscript.

Comment 1-13: P5: consider rephrasing/clarifying “However, constrained by the available passive analog RLC elements, the negative impedance attainable with analog non-Foster circuits may not span a continuous range, and their tuning often depends on tedious circuit component replacements. Therefore, analog non-Foster matching networks may not synthesize arbitrary FD, facing challenges with discretized parameter space and complicated manual tuning.” It seems the authors are trying to state that RLC components are commercially available only at discrete values, therefore constraining non-Foster circuits to corresponding discrete values.

Response 1-13: Many thanks for the comment.

Fig. R1 Negative impedance characteristics of non-Foster electronics.

In order to completely eliminate the effect of eddy current resistance and inductance on the transducer radiation bandwidth, the negative impedance characteristics attainable with non-foster circuit elements should have an arbitrary FD as shown in Fig. R1, i.e., the output behavior of non-foster circuit elements can be varied arbitrarily with operating frequency of electroacoustic transducer. There are two intractable issues that make it difficult to implement arbitrary FD for analog non-Foster circuit elements.

Firstly, yes, the discretized parameter space of analog non-Foster circuits. Due to the limited choice of available circuit components, the negative impedance attainable with analog non-Foster circuits are limited in their discretized parameter space and cannot span in an arbitrary range.

Secondly, tuning of the output negative impedance of analog non-Foster circuit elements is often accompanies by tedious circuit component replacements.

The load impedance Z is composed of passive analog RLC elements. Once part numbers and specifications of the active and passive components in the analog non-Foster matching networks have been determined, the terminal negative impedance value of the element are subsequently determined. The tuning of the terminal negative impedance value implies tedious circuit component replacements.

In sum up, analog non-Foster matching networks have difficulties in synthesizing arbitrary FD, facing challenges with discretized parameter space and complicated manual tuning. On the contrary, our proposed method replaces the passive analog RLC elements with equivalent reference negative resistor, negative inductor, or negative capacitor during the digital signal processing, thus achieving arbitrary frequency dispersion.

In order to avoid possible confusion, we have changed the related sentence, please refer to the **red words** on Section “Digital non-Foster-inspired electronics” on page 6, Line 5-11 of the revised manuscript.

Comment 1-14: P7 Eq (1): the description of variable VD does not say whether it is peak, peak-peak, rms, etc. Is it a complex phasor?

Response 1-14: We do apologize for the mistake in the formula in the original manuscript. In the equivalent circuit model for an electroacoustic transducer shown in Fig. R1, the volume velocity V_D is equivalent to the current in circuit. Since the impedance in the equivalent circuit has both real and imaginary parts, V_D is a complex phasor, as Eq. (R1) (Corresponding to Eq. (1) in the original manuscript)

$$P_{ar} = \frac{1}{4} |V_D|^2 \text{real}(Z_{rad})$$

R1

Over the low frequency range, the transducer is conceived to operate as a rigid piston mounted in a flat infinite baffle board. The acoustic power radiated to the front of the baffle can be calculated by Eq. (R1) [1]. Where V_D is volume velocity and Z_{rad} is the acoustic radiation impedance as shown in Fig. R1 (Corresponding to Fig. 2a in the original manuscript).

Fig. R1. The overall system impedance model diagram.

[1] Leach, W. M. Introduction to electroacoustics and audio amplifier design 85-105 (Kendall Hunt, 2010).

Comment 1-15: P7 Eq (1): units are not given for volume velocity V_D or impedance Z_{rad} or power P_{ar} . Please avoid the ambiguous impedance units of Rayl, since there are 2 different definitions in use. Please specify impedance units such as $N \cdot s/m^3$.

Response 1-15: Thank you for the reminder! In Eq. (R1) (Corresponding to Eq. (1) in the original manuscript), the unit of volume velocity V_D is $g/(s \cdot cm^2)$, the unit of acoustic radiation impedance Z_{rad} is $Pa \cdot s/m^3$, and the unit of acoustic radiation power P_{ar} is Watt (W). Additionally, we have carefully scrutinized the revised manuscript and add units to all defined variables throughout the paper.

$$P_{ar} = \frac{1}{4} |V_D|^2 \text{real}(Z_{rad})$$

R1

For the revision, please refer to the red words on Supplementary Section 1 on page 2-3 of the revised Supplementary Information.

Comment 1-16: Please check to add units to all defined variables throughout the paper.

Response 1-16: Thanks for your suggestion! have carefully scrutinized the revised manuscript and add units to all defined variables throughout the paper.

Comment 1-17: P7 Eq (2), in text below Eq 2 it would be helpful to just say $Z_D(w)=RD(w)+LD(w)$ if that is true,

Response 1-17: Many thanks for your suggestion. We fully agree. For the revision, please refer to the red words on Supplementary Section 1 on page 2-3 of the revised Supplementary Information.

Comment 1-18: P7 bottom: it is unclear which of the two preceding sentences “this” is referring to: “This results in a reduction in radiation efficiency and considerably limits the operational bandwidth”.

Response 1-18: Sorry for the confusion! “This” was intended to mean “the eddy current impedance’s increase with frequency. More eddy losses will further reduce the acoustic power. Eventually, the operational bandwidth of the transducer is constrained. The corresponding text has been changed.

In order to avoid possible confusion, we have changed the related sentence, please refer to the red words on Section “Theoretical analysis of digital non-Foster-inspired electronics” on page 9, Line 4 of the revised manuscript.

Comment 1-19: P10: u_0 and i_0 are not defined?

Response 1-19: Sorry for the unclarity. As shown in Fig. R1, u_0 and i_0 represent the terminal output voltage of our non-foster circuit and the current through the circuit element, respectively. We have enlarged the fonts in the figure.

Fig. R1. General control strategy of the digital non-Foster electronics.

In order to avoid possible confusion, we have changed the related sentence, please refer to the red words on Section “Implementation of digital non-Foster-inspired electronics” on page 11, Line 7 of the revised manuscript.

Comment 1-20: P10: where authors state “The operation frequency f_s is first acquired”, please comment on the acquisition time and the accuracy/precision of the estimated frequency. Also, would the accuracy of the frequency estimate set a lower bound on the closed-loop bandwidth and/ or on the resonator cutoff bandwidth set by ω_c ?

Response 1-20: This is an in-depth question! We have addressed it through additional results and analysis.

First of all, as stated by the reviewer, we agree the accuracy of the frequency estimation will set a lower bound on the overall system bandwidth (the instantaneous bandwidth for information transmission).

The conventional PR controller with the cutoff bandwidth ω_c around the resonant frequency can be expressed in the s-domain as:

$$G_{PR}(s) = K_p + K_r \frac{2\omega_c s}{s^2 + 2\omega_c s + \omega_0^2}$$

R1

where K_p and K_r are the proportional term and the integral term of the controller, respectively. $\omega_0 = 2\pi f_0$ is the fundamental angular frequency.

The PR controller can obtain a large gain within the cutoff bandwidth ω_c for zero sinusoidal steady-state error tracking so as to ensure that the proposed element achieves the desired negative impedance, but the gain drops sharply when the estimated frequency deviates from the range of the cutoff bandwidth. When the estimated frequency deviates from the cutoff bandwidth ω_c , small gain cannot realize the zero steady-state error of the sinusoidal reference signal. **In this case, the inadequate impedance matching will impair the SPL curve so that the bandwidth expansion for electroacoustic transducers will be undermined.**

Secondly, in the proposed switch-mode digital non-Foster circuit element, a voltage comparator and eCAP module of DSP 28335 are used for frequency detection.

Fig. R1. (a) The zero-crossing comparison implemented by voltage comparator. (b) Comparison of input and output signal of the voltage comparator

The zero-crossing comparison implemented by the voltage comparator is shown in Fig. R1(a). The signal u_i is inputted from the positive input pin and the negative input pin is grounded. When $u_i > 0$, the output voltage u_o is equal to the positive supply voltage U_{DC} , and when $u_i < 0$, the output voltage u_o is equal to zero, as shown in Eq. (R2). According to Eq. (R2), it is observed that when the input signal u_i is a sine signal, the output signal u_o is a square wave signal with the same frequency of u_i as shown in Fig. R1(b).

$$u_o = \begin{cases} U_{DC} & u_i > 0 \\ 0 & u_i < 0 \end{cases} \quad \text{R2}$$

The output pin of voltage comparator is connected with the eCAP module of DSP28335, which can capture the rising and falling edges of the square wave signal and thus calculate the width and duty cycle. Assuming that eCAP module captures the first rising edge at t_1 , followed by the falling edge at t_2 , the capture frequency f_{ecap} can be calculated from Eq. (R3).

$$f_{\text{ecap}} = \frac{1}{2(t_2 - t_1)} \quad \text{R3}$$

In order to calculate the operating frequency of the electroacoustic transducer, the eCAP module of DSP 28335 requires to capture at least one rising edge and one falling edge. Therefore, the acquisition time of frequency estimation takes no less than half a cycle.

In this manuscript, LM393 (Texas Instruments) with supply voltage of 3.3V is used as voltage comparator. In the actual process of frequency estimation, the output signal of the voltage comparator is a non-ideal square wave with rising and falling edges and the precision of the estimated frequency is mainly determined by the rise-fall times of output signal u_o . Taking into account the input voltage range, the driving capability and the rise-fall times, the values of the pull-up resistance R_{UP} , input terminal R_{I1} and R_{I2} , and feedback resistance R_f are selected as shown in Table R1 below.

Table R1 The selected resistance values of the voltage comparator.

Resistance	R_{UP}	R_{I1}	R_{I2}	R_f
Value/ Ω	10M	100K	100K	1K

Taking the operating frequencies f_s of the electroacoustic transducer at 500Hz and 1000Hz as an example, the input and output waveforms of voltage comparator were recorded by the oscilloscope (MPD34, Tektronix, Inc.) during the experiment as shown in Fig. R2.

(a)

(b)

Fig. R2. The input and output waveforms of voltage comparator with the zoomed-in graphs for the rise-fall times. (a) $f_s = 500\text{Hz}$. (b) $f_s = 1000\text{Hz}$.

The GPIO pins of DSP 28335 has a legal input high level range of 2.0V-3.3V, and a legal input low level range of 0V-0.8V [1]. Therefore, the transition process between the 2.0V high level and 0.8V low level is considered as the rising and falling edges of the output signal. According to Fig. R2, the rising time t_{up} and failing time t_{off} of the output non-ideal square wave signal are approximately 40ns and 20ns, respectively. In summary, the deviation of the estimated period is less than $0.12\mu\text{s}$ as shown in Eq. (R4).

$$\Delta t \leq 2(t_{up} + t_{off}) \approx 0.12\mu\text{s}$$

R4

The cutoff bandwidth of the proposed PR controller ω_c is set to 10π in this work, which is much larger than the frequency estimation error in the operating range (when $f_s = 2\text{kHz}$, $\Delta\omega \approx \pi \ll 10\pi$) of our proposed method according to Eq. (R5). In other words, below 2 kHz, the frequency estimation will have very little impact on the system performance.

$$\Delta\omega = 2\pi(f_{\text{ecap}} - f_s) = 2\pi \left(\frac{1}{2(t_2 - t_1)} - \frac{1}{2(t_2 - t_1) + \Delta t} \right) = \frac{2\pi\Delta t}{(2(t_2 - t_1) + \Delta t)(2(t_2 - t_1))}$$

R5

However, it can be clearly seen if f_s increases, its period $1/f_s$ will decrease. When $1/f_s$ approaches 10π , the frequency estimation will degrade very quickly, which will lead to the great deviations on the equivalent negative impedance by our approach. This will surely impair the SPL curve and finally, impair the bandwidth expansion for electroacoustic transducers.

For the revision, please refer to the **red words** on Supplementary Section 3 on page 9, Line 9-11 of the revised Supplementary Information.

[1] Cree. Datasheet of TMS320F28335 Real-Time Microcontrollers (2022).

Comment 1-21: P10: if the frequency being referred to in the following sentence is the operating frequency f_s , please say so: “a large gain at an arbitrary known frequency”.

Response 1-21: We agree with the reviewer and accept the comments. The word “**an arbitrary known frequency**” has been changed into “**an arbitrary operating frequency f_s** ” in the revised manuscript. For the revision, please refer to the **red words** on Section “Implementation of digital non-Foster-inspired electronics” on page 12, Line 6 of the revised manuscript.

Comment 1-22: P10: in the phrase “has a fast dynamic performance and stability” please state some specific quantities such as closed-loop bandwidth for speed, and such as gain/phase margins for stability.

Response 1-22: Many thanks for the suggestion! As Fig. R1 shows the closed-loop transfer function block diagram of our circuit, and the corresponding open-loop transfer function block diagram is as shown in Fig. R2.

Fig. R1. The closed-loop transfer function block diagram of the digital non-Foster electronics.

Fig. R2. The open-loop transfer function block diagram of the digital non-Foster electronics.

By simplifying the open-loop control block diagram, and the transfer function from the reference current I_{ref} to the output current I_o can be derived as

$$G_{\text{open}}(s) = -\frac{G_{\text{PR}}(s)G_{\text{PWM}}(s) + K_f G_{\text{PWM}}(s)}{s^3 L_s C_s L_T + s^2 L_s C_s R_T + s L_T + s L_s + R_T + K_f G_{\text{PWM}}(s)} \quad \text{R1}$$

Based on the open-loop transfer function, the closed-loop transfer function can be further derived as

$$G_{\text{closed}}(s) = \frac{G_{\text{open}}(s)}{1 + G_{\text{open}}(s)} \quad \text{R2}$$

where the filter capacitor C_s and the filter inductor L_s are respectively equal to 2μF and 2mH. R_T and L_T are respectively the resistance and inductance of the electroacoustic transducer. K_f is the feedforward gain. The transfer function of the PR controller is introduced

$$G_{\text{PR}}(s) = K_p + K_r \frac{2\omega_c s}{s^2 + 2\omega_c s + \omega_0^2} \quad \text{R3}$$

where ω_c represents the cutoff bandwidth around the resonant frequency ω_0 . Also, K_p and K_r are the proportional term and the integral term of the controller, respectively.

Moreover, $G_{\text{PWM}}(s)$ can be modeled as

$$G_{\text{PWM}}(s) = \frac{1 - 0.5T_s s}{(1 + 0.5T_s s)^2} \quad \text{R4}$$

where T_s is the sampling time, which is equal to 25μs in this work.

Here, we take the operating frequency of 900 Hz as an example to solve the gain/phase margins and closed-loop bandwidth of the proposed control system. The detailed parameter configurations of 900 Hz are shown in Table R1.

Table R1 The detailed parameter configuration of 900 Hz

Operating frequency f_s	K_f	K_p	K_r	ω_c	R_T	L_T
900 Hz	3	1.5	150	10π rad/s	7.40515 Ω	4.37 mH

Based on the open-loop transfer function shown in Eq. (R1) and the parameter configurations in Table R1, the open-loop bode plot of the control system in the operating frequency of 900 Hz can be obtained as shown in Fig. R3.

Fig. R3. The open-loop bode plot of the digital non-Foster electronics in the operating frequency of 900 Hz.

It can be seen from Fig. R3 that when the cutoff frequency ω_{co} is 5779 rad/s (at this point the open-loop gain is 0 dB), the phase margin is equal to 40.07° , which is bigger than 0° and is in the optimal interval between 30° - 60° . In addition, since the crossing frequency (at this point the phase angle is equal to -180°) is not near the resonance frequency, the gain margin is less than 0 dB.

Fig. R4. The closed-loop bode plot of the digital non-Foster electronics in the operating frequency of 900 Hz.

Based on the closed-loop transfer function shown in Eq. (R2) and the parameter configurations in Table R1, the closed-loop bode plot of the control system in the operating frequency of 900 Hz can be obtained as shown in Fig. R4. For a bode plot without DC gain, the closed-loop bandwidth is defined as the frequency range from peak gain to -3 dB. Therefore, it can be seen from Fig. R4 that the closed-loop bandwidth of the control system in 900 Hz is approximately equal to 20π rad/s.

Many thanks for your kind help! For the revision, please refer to the **red words** on Supplementary Section 2 on page 4-6 of the revised Supplementary Information.

Comment 1-23: P11: please also state the center frequency of the passive-tuned case in “with a radiation bandwidth of 49 Hz”.

Response 1-23: We are sorry for the incorrect description on the passive-tuned experimental procedure and results in Page 11 Line 19. The description “In Fig. 3e, the passive matching using conventional Foster components is shown. A total capacitance of 1000 μ F was used to impedance match, with a radiation bandwidth of 49 Hz.” should be corrected to “In Fig. 3e, the passive matching using conventional Foster components is shown. Here, the total capacitance value is equal to 1000 μ F, which is used to compensate the eddy current inductance of the acoustic radiator. The achieved resonant frequency is 50.98 Hz with a bandwidth of approximately 137 Hz.” We have supplemented the experimental details as follows. For the revision, please refer to the **red words** on Section “Experimental verification of digital non-Foster-inspired electronics” on page 14, Line 20-23 of the revised manuscript.

Comment 1-24: P13: if the “transient process lasts only two or three cycles” in estimating f_s and changing the resonance filter, would this imply that the instantaneous bandwidth of the proposed adaptive PR approach is approximately $f_s / 3$? If so, the reader should be cautioned that the bandwidth of the Fig 3e curve denoted “w/ FD” is not indicating instantaneous bandwidth, whereas the curve denoted “w/o FD” does actually represent an instantaneous bandwidth.

Response 1-24: Many thanks for the comment! We fully understand your concern.

1) Sound pressure level (SPL) curve of the transducer by our proposed.

The SPL curve shown in Fig. R1 was measured according to a typical way of extracting the bandwidth of acoustic projector [1-3]. A Tektronix AFG1062 function generator was used to provide approximately constant voltage output. The velocity of the transducer was measured at a central point on the radiating surface using a Keyence LK-H020 laser displacement sensor. The radiating surface supports predominately a longitudinal vibration motion (all points on the radiating surface moving in phase) in this frequency range with the highest levels of vibration seen along the central equator of the surface. The displacement sensor captures the magnitude of this motion by measuring the outward surface velocity.

Fig. R1 Comparison of the sound pressure level (SPL) curves under different impedance matching approaches and the measured vibration mode of the transducer using a laser sensor (inset figure).

The vibration velocity of the transducer was recorded by the laser displacement sensor and then converted into the corresponding SPL (see Supplementary Section 4 and Extended Data Fig. 4). By these steps, the SPLs at different frequency could be obtained and the SPL curve shown in Fig. R1 (Corresponding to Fig. 3e in the original manuscript) was depicted. In Fig. R1, the same experimental setup is used for w/o FD. The difference lies in the factor that without FD, the equivalent negative capacitor or inductance will not vary with frequency.

2) Analysis and reduction for transient response time

We do agree that it is essential to minimize the transient response time as much as possible, as pointed out by the reviewer. According to Shannon's theorem [4] as shown in Eq. (R1) (where C is the channel capacity, B is the channel bandwidth, S and N are respectively the average power of active signal and noise in channel), long transient times, which correspond to an increase in the noise signal and a decrease in the active signal, not only result in distortion of the modulated signal and loss of information, but also lead to a low data transmission rate. The issues caused are similar to narrow instantaneous bandwidths.

$$C = B \log_2 \left(1 + \frac{S}{N} \right)$$

R1

The transient process by our approach consists of three main steps: estimating operating frequency f_s , changing the resonance filter parameters, and controller transient response. Among them, the parameter calculation and adjustment process can be optimized by increasing the processing speed of microprocessor. Moreover, the transient response to achieve steady-state can also be optimized by the adjustment of the resonance filter parameters. Finally, the process of operating frequency estimation can be eliminated by the algorithm optimization in practical applications.

For example, one of the feasible methods is delayed synchronous running. When the first command code is received, the frequency estimation and parameter calculation are carried out first, and then waiting for the arrival of the next command code. When the second command code is received, the modulated signal of the previous command code is transmitted synchronously with negative impedance matching of the proposed switch-mode digital non-Foster circuit element. At this moment, owing to the perfectly known frequency and corresponding controller parameters, the transient process consists only of the dynamic response of the controller.

We have experimentally tested this idea. In this way, the proposed switch-mode digital non-Foster circuit element can ensure the synchronous operation with the signal source, which greatly reduce the transient response process of the proposed digital non-Foster electronic, as shown in Fig. R3.

Fig. R3 Transient response processes of the proposed digital non-foster circuit element by algorithmic optimization

For the revision, please refer to the **red words** on Supplementary Section 7 on page 13-14 of the revised Supplementary Information.

- [1] Rasmussen, C. & Alù, A. Non-Foster acoustic radiation from an active piezoelectric transducer. *Proc. Natl. Acad. Sci. USA*. **118**, e2024984118 (2021).
- [2] Butler, J. L. & Sherman, C. H. Transducers and arrays for underwater sound 517-553 (Cham, Switzerland:20 Springer International Publishing, 2016)
- [3] Leach, W. M. Introduction to electroacoustics and audio amplifier design 85-105 (Kendall Hunt, 2010).
- [4] Shannon, C. E. A mathematical theory of communication. *Bell System Tech. J.* **27**, 379-423 (1948).

Comment 1-25: Supplement P6: poles instead of roles in “the system roles will turn”.

Response 1-25: Sorry for this error. We have corrected it in the revised version. For the revision, please refer to the **red words** on Supplementary Section 2 on page 6, Line 9 of the revised Supplementary Information.

Reviewer #2:

Comments to the Author: This manuscript presents an interesting and certainly very useful method for increasing the radiated power of an originally mismatched radiator over a wide bandwidth. The results presented are correct and convincing and can be very important for practical engineering systems. There is also no doubt that the presented approach is far more flexible than the widely studied linear non-Foster matching and that it offers higher operating

power levels and better efficiency. Although the manuscript is based on some well-known ideas, I believe that the presented research results are novel enough to be published in Nature Communications.

In the present form of the manuscript, however, there are several serious terminological problems that may lead the reader to wrong conclusions. In addition, there are some previous representative studies with similar ideas that, at least in my opinion, should be included in the reference list. Finally, some important limitations of the presented approach are not emphasized. In light of all this, I recommend a comprehensive revision of this manuscript.

My specific comments are given below:

Response: We highly value the reviewer's careful read of our paper, as well as their insightful and encouraging remarks. We have followed their advice to make significant adjustments, particularly to the inadequate literature review and terminology errors.

Comment 2-1: 1) Terminology problems

I think that the proposed title ("Digital Non-Foster Electronics for Broadband Impedance Matching") is inappropriate. Foster's theorem states that the reactance and susceptance of a lossless network always increase with frequency. Of course, this theorem only applies to networks without resistors. Foster's theorem does not apply to lossy networks (the phenomenon of anomalous dispersion is a very well-known example). In other words, the Foster networks can only contain ordinary (positive) capacitors or/and inductors (or, in the distributed element scenario, the sections of lossless transmission lines). On this basis, the term non-Foster elements is used in the literature almost exclusively for electronic circuits that behave like negative capacitors and negative inductors, but (in ideal case) not like negative resistors. There are some authors who place negative resistors within the scope of non-Foster elements, but I think this is fundamentally wrong and causes unnecessary confusion. In this manuscript, the authors make extensive use of negative resistance, which means that in addition to reactance cancelation with negative inductance and negative capacitance, the system also has the characteristics of a negative-resistance reflection amplifier. The authors say that the generated negative resistance cancels the positive resistance, which physically means that it compensates the losses with the reflection gain based on negative resistance. Therefore, I firmly believe that using the term "non-Foster matching" in this context is incorrect. The authors may use "non-Foster-like matching" or "non-Foster-inspired matching" (the terminology used in some related studies [R1, R2] that use both non-Foster elements and negative resistances for amplification) or "generalized active broadband matching with arbitrary dispersion cancelation" or something similar, but not non-Foster matching. The authors should not only change the terminology related to "non-Foster", but also make corresponding changes throughout the manuscript (e.g., clearly explain that the use of a negative resistance causes reflection amplification that boosts radiated power).

[R1] A. Kiricenko and S. Hrabar, "Non-Foster-inspired Two-transmitter System based on Coherent Current/voltage Sources," 2022 IEEE International Symposium on Antennas and Propagation and USNC-URSI Radio Science Meeting (AP-S/URSI), Denver, CO, USA, 2022, pp. 1308-1309, doi: 10.1109/AP-S/USNC-URSI47032.2022.9886333

[R2] S. Hrabar, I. Cavlek, D. Mikulic, S. Milic and E. Sopp, "Extension of Non-Foster-inspired Two-transmitter Matching to Arbitrary Antenna Impedance," 2021 International Symposium ELMAR, Zadar, Croatia, 2021, pp. 49-52, doi: 10.1109/ELMAR52657.2021.9550906.

Response 2-1: Many thanks for your suggestion and the careful reading of our work. We understand your criticism, but partially disagree with it. The reviewer is correct that Foster's reactance theorem implies a positive derivative only for frequency regions with low-loss, and cannot be applied to highly absorptive frequency regions, since the frequency behavior can show a negative slope associated with anomalous dispersion. We clarify in the following our terminology.

1) Non-foster's theorem

Foster's theorem was first proposed in Ref. [1]. As the reviewer said, Foster's theorem, which is also known as reactance theorem, is applicable to lossless networks. The content of the theorem can be briefly formulated as a lossless one-port network, whose input reactance and susceptance slopes with frequency are both strictly monotonically increasing functions. Its mathematical expression is

$$\frac{\partial X(\omega)}{\partial \omega} > 0, \frac{\partial B(\omega)}{\partial \omega} > 0 \quad \text{R1}$$

Elements that satisfy Foster's theorem are called Foster elements, e.g., lossless capacitors and inductors, etc. On the contrary, as stated by the reviewer, a non-Foster network or a non-Foster element should be defined as a lossless one-port network [2], whose input reactance and susceptance slopes with frequency are both strictly monotonically decreasing functions. Its mathematical expression is

$$\frac{\partial X(\omega)}{\partial \omega} < 0, \frac{\partial B(\omega)}{\partial \omega} < 0 \quad \text{R2}$$

The region of negative slope in a passive network due to large absorption, also known as anomalous dispersion, does not violate the non-Foster theorem, and it can be achieved with passive lossy elements. A non-Foster element that obeys Eq.(R2) over a wide range of frequencies and with negligible absorption needs to have gain instead. In our paper, we employ a negative resistance to realize non-Foster elements, i.e., the terminal resistance of our circuit has a negative value as frequency varies, corresponding to gain. Given the active nature of our circuit, we believe that non-Foster matching can be a proper terminology, but inspired by this comment we changed it into 'non-Foster-inspired' to avoid confusion and to match the prior literature, specifically Ref. [3] and Ref. [4].

2) Negative resistance output characteristic

By self-adaptive PR approach, our proposed digital non-Foster-inspired circuit provides feedback control on the voltage-current output characteristics, which can exhibit an equivalent negative resistor. Although the implementation is distinct from that of the negative resistance amplifier, as described by the reviewer, the system has the same terminal characteristics of a negative-resistance reflection amplifier, which physically means that it compensates the losses of the electroacoustic transducer based on negative resistance output characteristics.

Fig. R1. (a) A negative resistance reflection amplifier realized by cross-coupled MOSFETs. (b) Simplified small signal schematic.

A negative resistance reflection amplifier realized by a MOSFET is shown in Fig. R1(a), and Fig. R1(b) is its small-signal model [5]. Based on the small-signal circuit model, the output characteristics of this negative resistance reflection amplifier R_{eg} can be given as:

$$R_{eg} = -\frac{2}{g_m} \quad \text{R3}$$

g_m represents the transconductance of MOSFET, which can be adjusted by varying the bias current I_{bias} . It can be seen from Eq. (R3) that the output negative resistance characteristics of the reflective amplifier are determined by the physical characteristics of devices (e.g., the output characteristics of the reflective amplifier are due to the transconductance g_m of MOSFETs in Fig. R1).

By comparison, our circuit realizes the output negative impedance characteristics via feedback close-loop control, which does not rely on the physical characteristics of MOSFET-like devices. The feedback control method implemented by the digital processing that produces a great control gain over a wide frequency range. Thus, “generalized active broadband matching with arbitrary dispersion cancelation” can be finally achieved.

In conclusion, we agree with the reviewer that our approach for simultaneous realization of equivalent negative resistance and negative reactance characteristics should be named as **non-Foster-inspired matching**. Furthermore, although the implementation is distinct from

that of a negative resistance amplifier, as described by the reviewer, the system has the same output characteristics of a negative-resistance reflection amplifier, which physically means that it compensates the losses of the electroacoustic transducer based on negative resistance output characteristics.

- [1] Foster, R. M. A reactance theorem. *Bell System Tech. J.* **3**, 259-267 (1924).
- [2] Sussman-Fort, S. E. & Rudish, R. M. Non-Foster impedance matching of electrically-small antennas. *IEEE Trans. Antennas and Propag.* **57**, 2230-2241 (2009).
- [3] Kiricenko, A. & Hrabar, S. Non-Foster-inspired Two-transmitter System based on Coherent Current/voltage Sources. In *2022 IEEE International Symposium on Antennas and Propagation and USNC-URSI Radio Science Meeting (AP-S/URSI)*, 1308-1309 (2022).
- [4] Hrabar, S. *et. al.* Extension of Non-Foster-inspired Two-transmitter Matching to Arbitrary Antenna Impedance. In *2021 International Symposium ELMAR*. 49-52 (2021).
- [5] Landsberg, N. & Socher, E. A low-power 28-nm CMOS FD-SOI reflection amplifier for an active F-band reflectarray. *IEEE Transactions on Microw. Theory Techn.* **65**, 3910-3921 (2017).

Comment 2-2: b) The term "digital non-foster" also seems confusing, as there are several published reports [R3, R4] that use this term for systems that in fact mimic ordinary non-foster systems with linear amplifiers with positive feedback but use a digital approach. In ordinary ("analog") non-Foster systems, the negative impedance is achieved by a superposition of the original signal and 'assisting signal' fed from the output of an amplifier via positive feedback. The "digital systems" from [R3, R4] first apply an A/D conversion, process the signal in the digital domain, apply a D/A conversion and again use a superposition. So, these systems really "imitate" classical "analog" non-Foster systems. In this manuscript, however, the authors actually use high-efficiency nonlinear switching systems that provide an assisting signal by low-pass filtering a discrete signal generated in an H-bridge with MOSFET switches. This is fundamentally different from the studies in the literature, but is similar to the patent in [R5]. Therefore, I believe that the term "switching mode" or "digitally controlled switching mode" or similar is much more appropriate than simply "digital". This would clearly differentiate this work from other studies in the literature.

[R3] T. P. Weldon, J. M. C. Covington, K. L. Smith and R. S. Adams, "Stability conditions for a digital discrete-time non-Foster circuit element," 2015 IEEE International Symposium on Antennas and Propagation & USNC/URSI National Radio Science Meeting, Vancouver, BC, Canada, 2015, pp. 71-72, doi: 10.1109/APS.2015.7304421.

[R4] D. M. Johnson and T. P. Weldon, "A Clock-Tuned Discrete-Time Negative Capacitor Implemented Using Analog Samplers," 2018 IEEE International Symposium on Circuits and Systems (ISCAS), Florence, Italy, 2018, pp. 1-5, doi: 10.1109/ISCAS.2018.8351121.

[R5] White et al 'Switched Mode Negative Inductor', US patent 9923548, May 2018

Response 2-2: Many thanks for the comment. Again, we do apologize for the insufficient literature review.

1) Prior-art digital non-Foster methods

The pioneering digital non-Foster circuit element was implemented by Dr Weldon (Ref. [1]) as shown in Fig. R1. **Open-loop control method** was used in this digital approach. The analog voltage at the input terminals $v(t)$ is first digitized with an ADC (analog-to-digital converter) into discrete-time signal $v[n]$. The behavior of the circuit is then established through calculating the appropriate current using a discrete-time filter $H(z)$ in the digital signal processing. Finally, the current at the input terminals $i(t)$ is converted from discrete-time current $i[n]$ by a current-output DAC (digital-to-analog converter).

Fig. R1. Block diagram of digital discrete-time non-Foster circuit.

Furthermore, Ref. [2] analyzed the stability of the digital discrete-time non-Foster circuit elements by judging the relative positions of poles of the corresponding discrete-time transfer function. Then, Ref. [3] and [4] discussed the possibility of **digital series negative RC (resistor-capacitor) and digital series negative RL (resistor-inductor) circuit** implementations by digital non-Foster circuit elements to improve stability or to mitigate parasitic resistance. Ref. [5] improves the accuracy and stability of the earlier digital non-Foster methods by refining the modeling of the digital non-Foster circuit elements and increasing the complexity of discrete-time filter $H(z)$. In Ref. [6], authors simulated the impedance matching using a non-Foster circuit element in series with the electrically-short monopole antenna model.

As the reviewer mentions, these digital non-Foster systems in fact mimic ordinary non-foster systems by linear analogue amplifiers with open-loop control by digital signal processing. In contrast to analog non-foster methods, these systems replace available passive analog RLC elements replacements with the adjustment of digital process parameters. Nevertheless, without the corresponding closed-loop control approaches, the practical realization of the prior-art digital non-Foster methods must take into account the purely technological issues (parasitics of the circuit layout and those of the attached devices, etc.), which will cause unexpected accuracy and stability issues.

2) Switch mode applied to non-Foster implementations

In Ref. [7], a switched mode negative inductance with a wide band performance was provided in the embodiment to allow impedance matching to high power levels. Our work is

distinguished from this patent in two aspects. **First, the switch-mode negative inductance implemented in this patent still relies on physical analog inductors and therefore, is still open-loop**, which cannot synthesize arbitrary values, equally facing challenges with discretized parameter space and complicated manual tuning. **Second, the square wave voltage must be used as an input signal**. If the input voltage is sinusoidal, the equivalent negative inductance will be difficult to find out, due to the introduced distortion. **As the reviewer pointed out, this patent lacks the implementation process in detail and experimental verification.**

By comparison with the **open-loop control methods** in the pioneering digital non-Foster circuit elements, we believe that our proposed **closed-loop**(self-adaptive PR approach) has two advantages. Firstly, our output negative capacitor or negative inductor or negative resistor is set **by a reference signal, which will not influence the controller setting**. **Secondly, our proposed approach does not rely on the modeling of prior-art digital non-Foster circuit elements or physical inductors/capacitors.**

Many thanks for your kind help! For the revision, please refer to the **red words** on Introduction on page 4, Line 6-12 of the revised manuscript.

- [1] Weldon, T. P., Covington, J. M., Smith, K. L. & Adams, R. S. Performance of digital discrete-time implementations of non-Foster circuit elements. *In 2015 IEEE International Symposium on Circuits and Systems (ISCAS)*, 2169-2172 (2015).
- [2] Weldon, T. P., Covington, J. M., Smith, K. L. & Adams, R. S. Stability conditions for a digital discrete-time non-Foster circuit element. *In 2015 IEEE Antennas and Propagation Society International Symposium (APSURSI)*, 71-72 (2015).
- [3] Kehoe, P. J., Steer, K. K. & Weldon, T. P. Thevenin forms of digital discrete-time non-Foster RC and RL circuits. *In 2016 IEEE Antennas and Propagation Society International Symposium (APSURSI)*, 191-192 (2016).
- [4] Kehoe, P. J., Steer, K. K. & Weldon, T. P. Stability analysis and measurement of RC and RL digital non-Foster circuits with latency. *In SoutheastCon 2017*, 1-4 (2017).
- [5] Daniel, C. G. & Weldon, T. P. A stable digital impedance circuit design method for resistive source impedances. *IEEE Open J. Circuits Syst.* **3**, 109-114 (2022).
- [6] Steer, K. K., Kehoe, P. J. & Weldon, T. P. Investigation of an adaptively-tuned digital non-Foster approach for impedance matching of electrically-small antennas. *In SoutheastCon 2017*, 1-5 (2017).
- [7] White et al 'Switched Mode Negative Inductor', US patent 9923548, May 2018

Comment 2-3: 2) Referencing problems

It seems that the authors mainly used references from the physics community. However, I believe that there are several very relevant papers from the engineering field that use similar ideas. For example, the concept of using additional sources that simultaneously implement non-Foster cancelation and negative resistance amplification (which is actually a linear, less efficient version of the authors' approach) was introduced in [R2]. In addition, an idea for a purely "digital" approach was presented in [R4]. A non-Foster approach with switching mode

and H-bridge was presented in [R5]. There are several other examples that can be found in the IEEEXplore database. In my opinion, the authors should pick up a few relevant engineering papers with similar ideas and include them in the reference list. I would like to be perfectly clear: it would simply be very good to include some previous studies with similar ideas, just to put the authors' work in context. I have gone through all these studies carefully and none of them went as far with experimental investigation and practical implementation. Therefore, I strongly believe that the authors' work is original enough to be published in Nature Communications (after implementing the suggested changes).

[R2] S. Hrabar, I. Cavlek, D. Mikulic, S. Milic and E. Sopp, "Extension of Non-Foster-inspired Two-transmitter Matching to Arbitrary Antenna Impedance," 2021 International Symposium ELMAR, Zadar, Croatia, 2021, pp. 49-52, doi: 10.1109/ELMAR52657.2021.9550906.

[R4] D. M. Johnson and T. P. Weldon, "A Clock-Tuned Discrete-Time Negative Capacitor Implemented Using Analog Samplers," 2018 IEEE International Symposium on Circuits and Systems (ISCAS), Florence, Italy, 2018, pp. 1-5, doi: 10.1109/ISCAS.2018.8351121.

[R5] White et al 'Switched Mode Negative Inductor', US patent 9923548, May 2018

Response 2-3: Many thanks for your support and guidance. We are sorry for our omission in the references. There was a lack of several very relevant papers from the engineering field, especially the development and application of the digital signal processing and the switch-mode in the field of non-Foster. Hence the additional references are given below:

The pioneering digital non-Foster circuits are proposed using open-loop control method, whose input port impedance is determined by the digital filter (Ref. [1]-[5]). In Ref. [6] and [7], the concept of using additional sources that simultaneously implement non-Foster cancelation and negative resistance was introduced. A high energy-injection switch-mode amplifier has been shown to realize a negative inductance [8] and negative resistance [9] [10].

Many thanks for your kind help! For the revision, please refer to the **red words** on Introduction on page 4, Line 6-12 of the revised manuscript.

[1] Weldon, T. P., Covington, J. M., Smith, K. L. & Adams, R. S. Performance of digital discrete-time implementations of non-Foster circuit elements. *In 2015 IEEE International Symposium on Circuits and Systems (ISCAS)*, 2169-2172 (2015).

[2] Weldon, T. P., Covington, J. M., Smith, K. L. & Adams, R. S. Stability conditions for a digital discrete-time non-Foster circuit element. *In 2015 IEEE Antennas and Propagation Society International Symposium (APSURSI)*, 71-72 (2015).

[3] Kehoe, P. J., Steer, K. K. & Weldon, T. P. Thevenin forms of digital discrete-time non-Foster RC and RL circuits. *In 2016 IEEE Antennas and Propagation Society International Symposium (APSURSI)*,

- 191-192 (2016).
- [4] Kehoe, P. J., Steer, K. K. & Weldon, T. P. Stability analysis and measurement of RC and RL digital non-Foster circuits with latency. *In SoutheastCon 2017*, 1-4 (2017).
 - [5] Daniel, C. G. & Weldon, T. P. A stable digital impedance circuit design method for resistive source impedances. *IEEE Open J. Circuits Syst.* **3**, 109-114 (2022).
 - [6] Hrabar, S., Cavlek, I., Mikulic, D., Milic, S. & Sopp, E. Extension of Non-Foster-inspired Two-transmitter Matching to Arbitrary Antenna Impedance. *2021 International Symposium ELMAR*, 49-52 (2021).
 - [7] Kirichenko, A. & Hrabar, S. Non-Foster-inspired Two-transmitter System based on Coherent Current/voltage Sources. *2022 IEEE International Symposium on Antennas and Propagation and USNC-URSI Radio Science Meeting (AP-S/URSI)*, 1308-1309 (2022).
 - [8] White et al 'Switched Mode Negative Inductor', US patent 9923548, May 2018
 - [9] Zhou, J., Zhang, B., Xiao, W., Qiu, D. & Chen, Y. Nonlinear parity-time-symmetric model for constant efficiency wireless power transfer: Application to a drone-in-flight wireless charging platform. *IEEE Trans. Ind. Electron.* **66**, 4097-4107 (2018).
 - [10] Assawaworrarit, S. & Fan, S. Robust and efficient wireless power transfer using a switch-mode implementation of a nonlinear parity-time symmetric circuit. *Nat. Electron.* **3**, 273-279 (2020).

Comment 2-4: 3) Lack of discussion on limitations of Authors' approach

The core idea of the authors' approach, which allows almost arbitrary frequency dispersion of the "synthesized" negative capacitance, inductance and resistance, is "on-the-fly" control by a digital PI regulator. Obviously, the highest operating frequency depends on the speed of a DSP control loop. The authors have clearly shown that this approach works in the audio range (up to 2 kHz). Is this the highest possible frequency? There are commercial digital shortwave transmitters with power amplifier/modulator that use MOSFET-based H-bridges up to 30 MHz. Would it be possible to apply the author's approach to these or even higher frequencies? I think it would be good to include a couple of sentences discussing the limitations imposed by available DSP technology.

Response 2-4: Many thanks for the inspiring question! Our proposed method is still limited by three other aspects for applications at higher frequencies.

Firstly, as you pointed out, the biggest limitation comes from the hardware of the chosen DSP chip. To ensure complete and accurate operation of each ISR, sampling time of DSP cannot be less than 25 μ s, i.e., the switching frequency should be below 40 kHz.

In this paper, we intentionally used Sine PWM technique, which is to build up a sinusoidal waveform with multiple pulses whose duty cycle vary sinusoidally with time (SPWM technique, over tens of pulses to form a low frequency sine). The reason for doing this is to achieve low harmonics [1]. Furthermore, the cutoff frequency of the LC filter used in the proposed digital non-Foster-inspired circuit is selected cautiously to be much less than the switching frequency f_{sw} so as to filter the higher harmonics out [2]. In this work, when $f_{sw}=40$ kHz, the cutoff frequency of the LC filter is limited to 4 kHz. As shown in Fig. R1, in

order to guarantee the quality of the output waveform of the proposed digital non-Foster-inspired circuit, the highest possible frequency cannot exceed 4 kHz (the output gain of the LC filter around this point is equal to -3 dB). In addition, the reduced number of adjustable pulses per operating period (f_{sw}/f_s) will also lead to longer transient response time of the controller. At low frequency applications, the transient response time is easy to constrict as for a low frequency period we have multiple pulses to adjust to achieve as fast as possible a desired sine shape. But at high frequencies, the number of adjustable pulses will be smaller, which means a longer transient response time.

Fig. R1. Schematic diagram of the limitation of the switching frequency on high frequency applications of digital non-Foster-inspired circuit. Red curve is harmonic content analysis of the output voltage of a H-bridge converter. f_{sw} is equal to 40 kHz. Blue curve is the output voltage gain schematic curve of LC filter. (a) $f_s = 4$ kHz. (b) $f_s = 10$ kHz.

There is, indeed, a high-frequency H-bridge converter, both legs of which are driven in a complementary fashion. Its output frequency is determined by the switching frequency so it can reach 30 MHz. Hence, digital signal processors with higher computing speeds (e.g. FPGA) are desirable for high-frequency applications. But in this case, the output of the H bridge will be rectangular, which contains high frequency harmonics. Inspired by the reviewer, we will further explore this high-frequency study in the future.

Secondly, as the reviewer said, the maximum switching frequency of MOSFETs is up to 30 MHz, but it cannot achieve high power due to the increased switching losses. The switching loss increases linearly with the switching frequency. So the typical SiC MOSFET rated at 1200 V switches below 100 kHz.

Thirdly, discretization errors. An essential step in the implementation of PR controller is the discretization. The proposed adaptive PR controller was discretized by Tustin method as Eq. (R1). The discretization error of PR controller is affected by two aspects: the sampling frequency and the discretization method. It can be seen from Fig. R2(a) that the deviation Δf_0 becomes more significant as the sampling time ($1/f_{sw}$) and f_s increase. Additionally, the discretization error cannot be eliminated with the change of the discretization method. It means a significant gain loss as shown in Fig. R2(b). Therefore, it is essential to employ the accurate discrete deviation compensation schemes to break the precision limit at high frequency applications. Otherwise, the desired output negative impedance cannot be accurately achieved, which would degrade the effect of impedance matching.

Fig. R2. (a) The frequency deviation Δf_0 from the resonance frequency of PR controller in s -domain f_0 (is equal to the operating frequency f_s) to the resonance frequency of PR controller in z -domain f_{d0} . f_{sw} is the sampling frequency that is equal to the switching frequency of SiC MOSFETs in the digital non-Foster circuit. (b) Schematic diagram of gain loss caused by frequency deviation of PR controller discretization.

For the revision, please refer to the **red words** on Supplementary Section 5 on page 10-11 of the revised Supplementary Information.

[1] Erickson, R. W. & Maksimovic, D. Fundamentals of power electronics (Springer Netherlands publishing, 2007).

[2] Wang, X., Loh, P. C. & Blaabjerg, F. Stability analysis and controller synthesis for single-loop voltage-controlled VSIs. *IEEE Tran. Power Electron.* **32**, 7394-7404 (2017).

Reviewer #3:

Comments to the Author: In this work, digital control techniques combined with switched mode circuits are applied to implement a non-Foster reconfigurable digital impedance matching network. As a result, the self-adaptive proportional resonant controller replaces the need for manual operation with in-situ tunability and improves power handling. In general, the developed of the paper is well conducted, as well as the results obtained. But there are some issues and observations that authors have to clarify and improve the manuscript.

Response: Many thanks for your support! We are very grateful for all the constructive suggestions. We have implemented a major correction as suggested.

Comment 3-1: The presentation of the captions of the Figures and their content should be expressed more clearly. As shown now they are very confusing. Please improve them.

Response 3-1: We are sorry for the unclarities. Now, most of the figures (5 of them are modified, 2 new figures are added) and content have been proofread. Some of figures are re-plotted as shown in the following, as well as the modified caption.

Fig. 3. Implementation of digital non-Foster-inspired electronics. **a.** General structure diagram of the digital non-Foster-inspired electronics. The proposed electronics consists of the signal stage and the power stage. In the signal stage, the power amplifier output voltage u_s and current i_o are converted by the sensors-based signal conditioning circuit into the voltage signals which meet the ADC (analog-digital conversion) input voltage range and ultimately are supplied to the DSP. The gate driver amplifies the DSP output signal and drives the action of H-bridge inverter. The amplitude-phase relation of the reference current i_{ref} and the reference output voltage u_{ref} of the proposed circuit is determined by setting the negative impedance reference Z_{ref} into the DSP. The power stage includes a DC source, a H-bridge with four SiC MOSFETs ($S_1 \sim S_4$), and a LC filter formed by inductor L_s and capacitor C_s . **b.** The control program flowchart in DSP. After initializing peripherals, the DSP remains in a waiting state until the eCAP interrupt service routine (ISR) or the timer ISR occurs. In the eCAP ISR, the DSP completes the operating frequency f_s calculation by detecting the rising edges and falling edges. The timer ISR mainly includes the reference current i_{ref} calculation, error calculation, self-adaptive PR closed-loop feedback control, and carrier comparison. **c.** Waveform schematic of the reference voltage u_{ref} , the reference current i_{ref} , and the output current i_o before self-adaptive PR closed-loop feedback control. **d.** The output signal $u_{S1} \sim u_{S4}$ of DSP, which is used to drive the behavior of the switch-mode electronics $S_1 \sim S_4$. **e.** Output voltage waveform schematic of H-bridge u_{AB} and LC filter u_o . **f.** Waveform schematic of u_{ref} , i_{ref} , and i_o after closed-loop control.

Many thanks for your kind help! For the revision, please refer to the **figures and contents**

of the revised manuscript.

Comment 3-2: Please show in a table the part number or model of the MOSFET, module driver, device used to implement DSP, current sensor, voltage sensor, inductors, capacitors, etc.

Response 3-2: We do apologize for the unclarity. Based on reviewer's comments, the part number or model of the main components as shown in Fig. R3 (corresponding to Fig. 3b in the original manuscript) and electroacoustic transducer have been listed in the table R1. And the table has been added in the revised manuscript.

Fig. R3. Photograph of the proposed digital non-Foster circuitry.

Table R1 The part numbers or values of the main components

Component	Part number or value	Producer
C_{AC}	350 V 10 μ F film capacitors	TDK Electronics AG, Germany
Current sensor I sensor	LA35-NP	LEM Electronics, Switzerland
Voltage sensor V sensor	TV16E	Dechang Electric Co., Ltd., China
DC/DC module	VRA2415ZP-10WR3	MORNSUN Technology Co., Ltd., China
	IB2403LS-1WR3	MORNSUN Technology Co., Ltd., China
ADC module	AD7656	Analog Devices, Inc., America
DSP	TMS320F28335	Texas Instruments, Inc., America
Module driver	UCC21520	Texas Instruments, Inc., America
SiC MOSFET	C3M0060065D	Wolfspeed, Inc., America
DC-link capacitor C_{DC}	450 V 820 μ F electrolytic capacitors	Nippon Chemi-Con Corp., Japan

	350 V 10 μ F film capacitors	TDK Electronics AG, Germany
Filter inductors L_s	2 mH	-
Filter capacitors C_s	630 V, 2 μ F film capacitor	KNSCHA Electronics Co., Ltd., China
Electroacoustic transducer	DL100LLB-01	Dongguan Huachuang Audio Equipment Co., Ltd, China

Many thanks for your kind help! The relevant content has been added in the revised Supplementary Information. Please refer to the **red words** on Extended Data Table 3 on page 29 of the revised Supplementary Information.

Comment 3-3: As a suggestion: The software program flowchart of the digital non-Foster circuit, shown in Extended Data Figure 3 can be added as very useful information to Figure 3 of the article document.

Response 3-3: Many thanks for the precious suggestion! We fully accept. As the reviewer stated, the software program flowchart of our system is modified and shown in Fig. R1 (corresponding to Extended Data Figure 3 in the Supplementary Information). It shows very useful information to demonstrating the timing of how our control functions. It has been added to the main manuscript as suggested. Meanwhile, we also keep Extended Data Figure 3 in the Supplementary Information as they are very detailed and helpful for experimental reproduction. (corresponding to Fig. 3(a) in the manuscript) contains some information from the software program flowchart in Fig. R1. Therefore, as suggested, we re-plotted Figure 3.

Fig. R1. Implementation of digital non-Foster-inspired electronics. a. General structure diagram of the digital non-Foster-inspired electronics. b. The control program flowchart in DSP. c~f. Waveform schematic of the implementation process of digital non-Foster-inspired electronics.

Many thanks for your kind help! For the revision, please refer to Fig. 3 and red words on Section “Implementation of digital non-Foster-inspired electronics” on page 11-13 of the revised manuscript.

Comment 3-4: In special I have some observations about Extended Data Section Figure 2 (Stability analysis of the digital non-Foster electronics):

In control systems, it is typically used to represent a signal that is fed back to two different summing points as two independent loops and not in the form presented in diagrams a, b, and c, where a single loop has two outputs to the summing point. Please correct.

Some of the simplifications shown in b, c, d, and e based on block algebra are correct, but the same result could be reached in a simpler way by applying the properties of feedback in summing points to the simplification of loops. The block diagram shown in figure f also can be simplified because the functions of input branches are connected in parallel as a sum. Please express the final result of the block diagram of transfer functions using the property of linearity of the system.

(Reference of the latter two comments: Feedback Control Systems, John van De Vegte, Prentice Hall, Third Edition, Chapter 3)

Response 3-4: Many thanks for your criticism! Yes, the previous presentation of the control block diagram was not rigorous and the simplifications in Extended Data Section Figure 2 (Stability analysis of the digital non-Foster electronics) in the original text are too intricate. We do apologize for this confusion.

According to your suggestion, we represent the output signal $I_o(s)$ that is fed back to two different summing points as two independent loops as shown in Fig. R1 (as shown by the red line) instead of a single loop which has two outputs to the summing point as shown in the original text.

Fig. R1. The transfer function block diagram of the digital non-Foster electronics.

Similarly, according to your suggestion, in order to obtain the transfer function of the our system in a simpler way, we simplify the transfer function block diagram in Fig. R1 by Mason’s gain formula as suggested [1], which applies the properties of feedback in summing

points to the simplification of loops. The detailed simplification process for a dual-input and single-output system is as follows.

Assuming $U_s(s)=0$, there are three touching loops and two possible forward paths from input node $I_{\text{ref}}(s)$ to output node $I_o(s)$ as shown in Fig. R1. The path gains P_1 and P_2 and the loop gains L_1 , L_2 , and L_3 are

$$P_1 = -\frac{G_{\text{PR}}(s)G_{\text{PWM}}(s)}{s^2L_sC_s(R_T + sL_T)} \quad P_2 = -\frac{K_f G_{\text{PWM}}(s)}{s^2L_sC_s(R_T + sL_T)} \quad \text{R1}$$

$$L_1 = \frac{G_{\text{PR}}(s)G_{\text{PWM}}(s)}{s^2L_sC_s(R_T + sL_T)} \quad L_2 = -\frac{1}{sC_s(R_T + sL_T)} \quad L_3 = -\frac{1}{s^2L_sC_s} \quad \text{R2}$$

K_f is the feedforward gain, which is added to handle the conflict between steady-state response, transient response, and stability [2]. Additionally, the transfer function of the PR controller is introduced [3]

$$G_{\text{PR}}(s) = K_p + K_r \frac{2\omega_c s}{s^2 + 2\omega_c s + \omega_0^2} \quad \text{R3}$$

where ω_c represents the cutoff bandwidth around the resonant frequency ω_0 . Also, K_p and K_r are the proportional term and the integral term of the controller, respectively. K_p ensures good transient performance and stability, while K_r can eliminate the amplitude and phase steady-state errors [4].

Moreover, $G_{\text{PWM}}(s)$ can be modeled as [5]

$$G_{\text{PWM}}(s) = \frac{1 - 0.5T_s s}{(1 + 0.5T_s s)^2} \quad \text{R4}$$

where T_s is the sampling time.

Mason's formula for the overall gain when $U_s(s)=0$ is

$$G_{\text{Iref}}(s) = \frac{I_o(s)}{I_{\text{ref}}(s)} = \frac{P_1\Delta_1 + P_2\Delta_2}{\Delta} \quad \text{R5}$$

where $\Delta=1-$ (sum of all individual loop gains) + (sum of products of gains of all possible combinations of two non-touching loops) - (sum of products of gains of all possible combinations of three non-touching loops) + ..., and Δ_k ($k=1,2$) is the cofactor of k th forward path, obtained from Δ by removing the loops that touch P_k path. Here, Δ and Δ_k respectively are

$$\Delta = 1 - L_1 - L_2 - L_3 \quad \Delta_1 = 1 \quad \Delta_2 = 1 \quad \text{R6}$$

Likewise, assuming $I_{\text{ref}}(s)=0$, there are three same touching loops and one forward path from input node $U_s(s)$ to output node $I_o(s)$ as shown in Fig. R1. The path gain P_3 and corresponding cofactor Δ_3 are

$$P_3 = \frac{1}{(R_T + sL_T)} \quad \Delta_3 = 1 - L_3 \quad \text{R7}$$

Mason's formula for the overall gain when $I_{\text{ref}}(s)=0$ is

$$G_{U_s}(s) = \frac{I_o(s)}{U_s(s)} = \frac{P_3 \Delta_3}{\Delta} \quad \text{R8}$$

Based on the Mason's gain formula, we simplify the transfer function block diagram of the digital non-Foster electronics as shown in Fig. R2.

Fig. R2. The final simplification result of transfer function block diagram of the digital non-Foster electronics.

According to Fig. R2, the formulation of the output signal $I_o(s)$ under the action of dual inputs $I_{\text{ref}}(s)$ and $U_s(s)$ can be derived further

$$\begin{aligned} I_o(s) &= G_{\text{lref}}(s) I_{\text{ref}}(s) + G_{U_s}(s) U_s(s) \\ &= \frac{P_1 \Delta_1 + P_2 \Delta_2}{\Delta} \cdot I_{\text{ref}}(s) + \frac{P_3 \Delta_3}{\Delta} \cdot U_s(s) \\ &= \frac{(P_1 \Delta_1 + P_2 \Delta_2) \cdot I_{\text{ref}}(s) + P_3 \Delta_3 \cdot U_s(s)}{1 - L_1 - L_2 - L_3} \end{aligned} \quad \text{R9}$$

To sum up, by Mersén's gain formula, we obtain the final simplification result of the transfer function block diagram of the digital non-Foster electronics in a simpler way. Thank you! For the revisions in the Supplementary Information, please refer to the highlighted lines in **red** on page 4-6. Previous Extended Data Section Figure 2 (Stability analysis of the digital non-Foster electronics) are all re-plotted as shown in page 18-19. Thank you very much!

- [1] Van De Vegte, J. Feedback Control System, 3rd ed. 85-89 (Englewood Cliffs, NJ: Prentice-Hall Publishing, 1994).
- [2] Li, Y. W., Loh, P. C., Blaabjerg, F. & Vilathgamuwa, D. M. Investigation and improvement of transient response of DVR at medium voltage level. *IEEE Trans. Ind. Appl.* **43**, 1309-1319 (2007).
- [3] Li, H. *et al.* A time-domain stability analysis method for grid-connected inverter with PR control based on floquet theory. *IEEE Trans. Ind. Electron.* **68**, 11125-11134 (2021).
- [4] Vidal, A. *et al.* Assessment and optimization of the transient response of proportional-resonant current controllers for distributed power generation systems. *IEEE Trans. on Ind. Electron.* **60**, 1367-1383 (2012).
- [5] Ye, T. *et al.* Analysis, design, and implementation of a quasi-proportional-resonant controller for a multifunctional capacitive-coupling grid-connected inverter. *IEEE Trans. Ind. Appl.* **52**, 4269-4280 (2016).

In all, we believe that a major correction following the reviewer's professional guidance is completed. We do hope that all the modifications are satisfactory!

With best wishes,

Prof. Xin Yang & Prof. Andrea Alù
Hunan University, China City University of New York, USA
On behalf of all authors

REVIEWERS' COMMENTS

Reviewer #1 (Remarks to the Author):

The revised manuscript has adequately addressed all earlier suggestions. The revised paper makes important novel contributions to the field.

Reviewer #2 (Remarks to the Author):

I have carefully studied the revised version and found that the authors have made all the necessary corrections and explained all the disputed issues in detail. In my opinion, this manuscript is now self-consistent and, definitely, much better. Therefore, I recommend the publishing of the manuscript in its present form.

Reviewer #3 (Remarks to the Author):

Dear authors:

All my comments and requests were answered clearly and your manuscript, in my opinion, is ready to be published.

Reviewer #3 (Remarks on code availability):

The revised code includes valuable information on the physical structure diagram as well as its representation from a control system point of view.

Response to The Comments

Manuscript ID: NCOMMS-23-41169A

Title: Digital Non-Foster-Inspired Electronics for Broadband Impedance Matching

Dear Reviewers,

We greatly appreciate your guidance and help on our manuscript. In addition, we have also provided a point-by-point response to the reviewers' comments as required.

Reviewer #1:

Comments to the Author: The revised manuscript has adequately addressed all earlier suggestions. The revised paper makes important novel contributions to the field.

Response: We are again sincerely grateful for your help, support and guidance! Thanks to your constructive and insightful comments, the paper can be considerably improved! Specially, confusions can be successfully avoided. We do believe that your professional suggestions will be of great significance to guide our future research. Thank you very much!

Reviewer #2:

Comments to the Author: I have carefully studied the revised version and found that the authors have made all the necessary corrections and explained all the disputed issues in detail. In my opinion, this manuscript is now self-consistent and, definitely, much better. Therefore, I recommend the publishing of the manuscript in its present form.

Response: We sincerely appreciate your support and guidance! We are very grateful for your time and effort to help us ensure that all the disputed issues have been well coped with, which would be very definitely essential for our work. With your important and invaluable suggestions, we do realize that there are more important opportunities for development of different digital non-foster-like circuits. We will further explore this area and very hope to have your guidance! Thank you very much!

Reviewer #3:

Comments to the Author: Dear authors: All my comments and requests were answered clearly and your manuscript, in my opinion, is ready to be published.

The revised code includes valuable information on the physical structure diagram as well as its representation from a control system point of view.

Response: We are sincerely grateful for your help and guidance! Your precious and professional suggestions on how to efficiently derive the transfer function block diagram is highly appreciated and very enlightening, which is very crucial to clarify our work. We also appreciate your time on helping us check the code. We learned a lot from you, which will be very important for our future work. Thank you very much!

With best wishes,

Prof. Xin Yang & Prof. Andrea Alù
Hunan University, China City University of New York, USA
On behalf of all authors